# Heterogeneous ice nucleation ability of aerosol particles generated from Arctic sea surface microlayer and surface seawater samples at cirrus temperatures

Robert Wagner[1], Luisa Ickes[2], Allan K. Bertram[3], Nora Els[4], Elena Gorokhova[5], Ottmar Möhler[1], Benjamin J. Murray[6], Nsikanabasi Silas Umo[1], and Matthew E. Salter[5]

[1]Institute of Meteorology and Climate Research, Karlsruhe Institute of Technology, Karlsruhe, Germany
[2]Department of Space, Earth and Environment, Chalmers, Gothenburg, Sweden
[3]Department of Chemistry, University of British Columbia, Vancouver, Canada
[4]Department of Ecology, University of Innsbruck, Innsbruck, Austria
[5]Department of Environmental Science and Analytical Chemistry & Bolin Centre for Climate Studies, Stockholm University, Stockholm, Sweden
[6]School of Earth and Environment, University of Leeds, Leeds, United Kingdom

*Correspondence to*: Robert Wagner (robert.wagner2@kit.edu)

**Abstract**. Sea spray aerosol particles are a recognised type of ice-nucleating particles under mixed-phase cloud conditions. Entities that are responsible for the heterogeneous ice nucleation ability include intact or fragmented cells of marine microorganisms as well as organic matter released by cell exudation. Only a small fraction of sea spray aerosol is transported to the upper troposphere, but there are indications from mass-spectrometric analyses of the residuals of sublimated cirrus particles that sea salt could also contribute to heterogeneous ice nucleation under cirrus conditions. Experimental studies on the heterogeneous ice nucleation ability of sea spray aerosol particles and their proxies at temperatures below 235 K are still scarce. In our article, we summarise previous measurements and present a new set of ice nucleation experiments at cirrus temperatures with particles generated from sea surface microlayer and surface seawater samples collected in three different regions of the Arctic and from a laboratory-grown diatom culture (*Skeletonema marinoi*). The particles were suspended in the AIDA cloud chamber and ice formation was induced by expansion cooling. We confirmed that under cirrus conditions, apart from the ice-nucleating entities mentioned above, also crystalline inorganic salt constituents can contribute to heterogeneous ice formation. This takes place at temperatures below 220 K, where we observed in all experiments a strong immersion freezing mode due to the only partially deliquesced inorganic salts. The inferred ice nucleation active surface site densities for this nucleation mode reached a maximum of about $5 \cdot 10^{10}$ m$^{-2}$ at an ice saturation ratio of 1.3. Much smaller densities in the range of $10^8 – 10^9$ m$^{-2}$ were observed at temperatures between 220 and 235 K, where the inorganic salts fully deliquesced and only the organic matter and/or algal cells and cell debris could contribute to heterogeneous ice formation. These values are two orders of magnitude smaller than those previously reported for particles generated from microlayer suspensions collected in temperate and subtropical zones. While this difference might simply underline the strong variability of the amount of ice-nucleating entities in the sea surface microlayer across different geographical regions, we also discuss how instrumental

parameters like the aerosolisation method and the ice-nucleation measurement technique might affect the comparability of the results amongst different studies.

## 1 Introduction

### 1.1 Sea spray aerosol as a source of ice-nucleating particles

A wealth of recent studies has substantiated early findings from the 1970s that sea spray aerosol (SSA) particles are able to act as ice-nucleating particles (INPs) in the immersion freezing mode for clouds in the mixed-phase temperature regime between 273 and 235 K (e.g. Bigg, 1973; Schnell and Vali, 1975; Knopf et al., 2011; Wilson et al., 2015; DeMott et al., 2016; Ladino et al., 2016; McCluskey et al., 2017; Creamean et al., 2019; Irish et al., 2019; Gong et al., 2020; Welti et al., 2020; Wilbourn et al., 2020; Wolf et al., 2020). As a comprehensive overview of previous investigations, we recommend the compilation in Table 1 by Ickes et al. (2020). Immersion freezing in the mixed-phase cloud temperature regime means that the SSA particles, once activated to cloud droplets, initiate the heterogeneous nucleation of ice crystals at lesser supercooling than required for the homogeneous freezing of pure water droplets, which takes place below about 235 K (Koop et al., 2000b). The heterogeneous ice nucleation ability is commonly attributed to the organic material contained in the SSA particles. Phytoplankton organisms, including algae, cyanobacteria, and fungi, are the major sources of organic matter in the ocean (Thornton, 2014; O'Dowd et al., 2015; Middelburg, 2019). Organic material is often enriched at the thin boundary layer between the ocean and the atmosphere, called the sea surface microlayer (Cunliffe et al., 2013; Zäncker et al., 2017), and released into the particle phase by the bursting of bubbles generated by breaking waves (Blanchard, 1964, 1989; Leck and Bigg, 2005). Submicron-sized film droplets, resulting from the fragmentation of the bubble membrane, have been shown to have particularly high mass fractions of organic matter (O'Dowd et al., 2004; Ault et al., 2013; Prather et al., 2013). Apart from this marine organic material, it has recently been proposed that residues from terrestrial dust deposited in seawater could be ejected by bubble bursting and thereby represent another, so far unrecognised ice-nucleating entity in SSA particles (Cornwell et al., 2020).

Previous ice nucleation measurements with SSA particles and their proxies featured a broad variety of approaches. They included stationary field studies at coastal sites (e.g. Mason et al., 2015; Ladino et al., 2019; Wex et al., 2019), ship- and aircraft-based measurements (e.g. McCluskey et al., 2018a; Hartmann et al., 2020; Welti et al., 2020), as well as complex laboratory experiments where phytoplankton blooms were simulated in large seawater tanks and wave channels, generating SSA particles by plunging sheets of water or breaking waves (e.g. DeMott et al., 2016; McCluskey et al., 2017; McCluskey et al., 2018b). Some experiments have specifically targeted phytoplankton organisms and their exudates (e.g. Alpert et al., 2011b; Knopf et al., 2011; Ladino et al., 2016; Tesson and Šantl-Temkiv, 2018; Wolf et al., 2019; Ickes et al., 2020), asking whether the ice nucleation behaviour of these species is representative of that observed for ambient sea surface microlayer samples. The two ice nucleation measurement techniques most frequently employed in the literature are droplet freezing assays and

continuous flow diffusion chambers (CFDCs), and the freezing data are usually reported as the temperature-dependent number of INPs per either droplet volume or volume of collected air. For ice nucleation measurements under cirrus conditions (see Sect. 1.2), INP concentrations are reported as a function of temperature and relative humidity. To quantitatively compare the ice nucleation ability of the SSA particles with that of particles from terrestrial sources like mineral dust, some measurements have also been analysed within the concept of the ice nucleation active surface site density, $n_s$, where the number of INPs is related to the dry aerosol surface area (Wilson et al., 2015; DeMott et al., 2016; McCluskey et al., 2017; Ickes et al., 2020). One important finding from these immersion freezing studies in the mixed-phase cloud temperature regime is that at least two different ice-nucleating entities in the SSA particles have to be considered, namely dissolved organic carbon INPs composed of ice-active molecules and particulate organic carbon INPs including ice-active cells or cell fragments of marine microorganisms (McCluskey et al., 2018b). In the course of this article, we refer to other specific results from these mixed-phase cloud studies. However, we first want to introduce the topic of our new measurements, which is the ice nucleation behaviour of SSA particles at cirrus temperatures below 235 K.

The atmospheric concentration of sea salt exhibits a strong decrease with altitude, from typically $0.3 - 3$ μg/m$^3$ in the boundary layer to less than 10 ng/m$^3$ above 6 km (Murphy et al., 2019). However, there is occasional evidence that sea salt particles and airborne microorganisms can be transported to the upper troposphere by deep convection (Ikegami et al., 1994; DeLeon-Rodriguez et al., 2013). Aircraft measurements focussing on the mass-spectrometric analysis of the residual particles from cirrus crystals showed contributions of up to 25% from sea salt over ocean and coastal regions (Cziczo et al., 2013). In these situations, mineral dust was the other main component of the ice residuals, suggesting a heterogeneous ice nucleation pathway for the formation of the cirrus crystals. Even if the global relevance of SSA particles as INPs in the cirrus regime turned out to be small, detailed experimental studies on this subject remain worthwhile if we are to explain regional importance of heterogeneous ice nucleation activity involving sea salt. The heterogeneous ice nucleation ability of sea salt particles in the cirrus region has also been considered as an explanation for the continued removal of sea salt with altitude even at very low temperatures where all cloud particles consisted of ice (Murphy et al., 2019). Moreover, from a mechanistic point of view, it is interesting to compare the heterogeneous ice nucleation ability of the SSA particles at both mixed-phase cloud and cirrus conditions. As such, some relevant questions are: 1) Do we see indications for multiple biogenic ice-nucleating entities in the SSA particles during the nucleation experiments at cirrus temperatures? 2) Do sea surface microlayer samples that are particularly ice-active at mixed-phase cloud conditions also show a superior ice nucleation ability at cirrus temperatures? In contrast to the large quantity of ice nucleation measurements at mixed-phase cloud temperatures, much less data is available that describe the heterogeneous ice nucleation ability of SSA particles and their proxies at cirrus temperatures. In the following section, we present a detailed summary of previous measurements to put our own new laboratory experiments and results into a broader context. These former studies are listed in Table 1.

## 1.2 A summary of ice nucleation measurements with SSA particles and their proxies at cirrus conditions

Cirrus clouds can either be formed in situ or originate from other cloud systems like deep convective clouds, where the ice phase is mostly formed through the supercooled liquid phase (Krämer et al., 2016). We do not consider the latter here, but focus on in situ cirrus, where ice crystals are directly formed at $T < 235$ K. Under these conditions, heterogeneous ice formation can proceed via different pathways. Ice nucleation from the supersaturated vapour on the surface of a solid or highly-viscous INP without prior formation of bulk liquid water is called deposition nucleation (Vali et al., 2015). Alternatively, heterogeneous ice nucleation is referred to as immersion freezing when ice-nucleating entities are immersed in supercooled aqueous solution droplets and cause their freezing at lower supersaturation or supercooling compared to the homogeneous freezing conditions of the pure solution droplets (Koop et al., 2000b). Note, immersion freezing in this case is different to immersion freezing under mixed-phase cloud conditions since it occurs below liquid water saturation. Also note that in recent years, there has been renewed evidence that in some cases, the deposition nucleation pathway is better described by the pore condensation and freezing (PCF) mechanism (Fukuta, 1966; Marcolli, 2014; Campbell et al., 2017; David et al., 2019). The PCF mechanism involves the condensation of microscopic quantities of water at suitable cracks or pores, which then goes on to freeze. However, we will continue to use the term "deposition nucleation" on some occasions to be consistent with cited literature.

In order to evaluate the available data, a distinction has to be made between the experimental approach and the initial phase state of the investigated particles. Three techniques have previously been adopted in studies that specifically investigated the heterogeneous ice nucleation behaviour of SSA particles and their proxies at cirrus conditions. The first type of studies involved deposition nucleation experiments with cells or cell fragments of the diatom *Thalassiosira pseudonana* using a temperature- and humidity-controlled ice nucleation cell, where the particles were deposited onto silicon wafer substrates (Knopf et al., 2011). The authors observed that the intact and fragmented diatoms acted as depositional INPs at temperatures below 240 K. The reported average ice nucleation onsets, as expressed in terms of $S_{ice}$, the saturation ratio with respect to ice, were in the range from 1.31 to 1.49 at temperatures between 236 and 205 K (Alpert et al., 2011a). In the second type of experiments, the same diatom species were probed in the immersion freezing mode, meaning that the *Thalassiosira pseudonana* cells or cell fragments were contained within aqueous NaCl droplets (Alpert et al., 2011b; Knopf et al., 2011). Here, the authors used an environmental cell where $40 - 70$ μm sized droplets that on average contained three diatom cells were deposited on hydrophobic glass slides, equilibrated to a certain relative humidity (RH), sealed against the environment, and finally cooled until freezing was detected with an optical microscope. The composition of the NaCl solution droplets remained constant during cooling because the amount of water vapour in the environmental cell was negligible compared to the amount of liquid water in the droplets. In reference experiments with pure NaCl solution droplets, ice nucleation onsets for homogeneous freezing were found to be in the $S_{ice}$ range from 1.46 to 1.51 at temperatures between 234 and 217 K. In the same temperature range, NaCl droplets containing *Thalassiosira pseudonana* cells or cell fragments nucleated ice at lower saturation ratios from 1.34 to 1.38 due to immersion freezing. The heterogeneous freezing data were parameterised by a horizontal shift of the ice melting curve on the water activity scale (Koop and Zobrist, 2009). It was later shown that the so-derived heterogeneous

freezing curve for *Thalassiosira pseudonana* is not only representative for the diatom's intact and broken cells but also for its exudates, because similar ice nucleation onsets were measured after filtering the diatom culture through a 0.1 μm filter (Wilson

et al., 2015). As a follow-up of their original ice nucleation study with *Thalassiosira pseudonana*, the authors probed the deposition nucleation and immersion freezing ability of two other marine algae, namely the green alga *Nannochloris atomus* and the coccolithophore *Emiliania huxleyi* (Alpert et al., 2011a). In the immersion freezing experiments, *Nannochloris atomus* behaved similarly to *Thalassiosira pseudonana*, showing $S_{ice}$ nucleation onsets between 1.36 and 1.42 for temperatures between 234 and 209 K, whereas the freezing onsets for *Emiliania huxleyi* were invariant compared to the homogeneous

freezing results for pure NaCl solution droplets. The variability in the ice nucleation behaviour was attributed to differences in the cell wall structure and chemical composition between the algal species (Alpert et al., 2011a). Whereas diatoms and green algae have polysaccharide coatings on the outer cell wall structures, there are no such structures in coccolithophores. Therefore, *Emiliania huxleyi* cells do not have coatings or layers of potentially ice-active organic material, rather the cells are covered by calcium carbonate plates. However, in the deposition nucleation experiments, where other morphological parameters of cell

walls, such as surface outgrowth, cracks, and pores, might control the heterogeneous ice nucleation efficiency (Marcolli, 2014; David et al., 2019), all phytoplankton types including *Emiliania huxleyi* showed similar average ice nucleation onsets as already specified above for *Thalassiosira pseudonana*.

In the third type of experiments, the SSA particles' ice nucleation ability was probed with CFDCs (Wilson et al., 2015; Ladino et al., 2016; Kong et al., 2018; Wolf et al., 2019; Wolf et al., 2020). In all of these studies, particles generated from either sea

surface microlayer samples or cultures of phytoplankton and marine bacterial species in seawater were first dried to RH ≤ 15% before entering the ice supersaturated region in the CFDC, thereby inducing efflorescence of the inorganic salt constituents (Koop et al., 2000a). This is important because at cirrus temperatures, crystalline inorganic salts can also be active as INPs. It is well known that for pure crystalline salts like ammonium sulphate, ammonium nitrate, and sodium chloride, there is a temperature-dependent competition between deliquescence and heterogeneous ice nucleation (e.g. Braban et al., 2001; Shilling

et al., 2006; Eastwood et al., 2009; Wise et al., 2012; Ladino et al., 2016; Wagner et al., 2020). Above around 225 K, the crystalline salt particles first deliquesce and nucleate ice by homogeneous freezing at $S_{ice} \geq 1.45$. Below that temperature threshold, however, they become active as INPs in the deposition nucleation mode before reaching the deliquescence point. In the case of SSA particles that contain not just a single but a mixture of inorganic salts, deliquescence is a gradual process. These particles go through partially dissolved states before finally becoming homogeneous aqueous solution droplets. They

begin to deliquesce at a low RH due to the presence of Ca and Mg salts with low deliquescence points, but only at about 74% RH (298 K), all of the remaining NaCl is dissolved and the particles transform to homogeneous droplets (Tang et al., 1997). Microscope images showed that the particle habit of an initially dry, irregularly-shaped sea salt particle changed to an overall spherical outline as the humidity was increased to $S_{ice} = 1$ at 215 K (Schill and Tolbert, 2014). At these conditions, the particle consisted of a brine layer of highly dissolvable salt components like $MgCl_2$ and $KMgCl_3 \cdot 6H_2O$ around a solid core of yet

undissolved salts, predominantly NaCl as the main component of sea salt. The two potential ice nucleation pathways for such

an internally mixed solid-liquid particle upon further increase of $S_{ice}$ are, (i) full particle deliquescence followed by homogeneous freezing, or (ii), immersion freezing due to the undissolved crystalline core before reaching the point of full deliquescence. The second process has also been termed as "deliquescent-heterogeneous freezing" in the literature (Khvorostyanov and Curry, 2004). Three independent laboratory studies have shown that below a threshold temperature of about 220 K the partly deliquesced SSA particles became active as INPs in the immersion freezing mode (Schill and Tolbert, 2014; Kong et al., 2018; Wagner et al., 2018). Specifically, the temperature-dependent ice nucleation measurements in the Aerosol Interaction and Dynamics in the Atmosphere (AIDA) cloud chamber yielded $S_{ice}$ freezing onsets between 1.24 and 1.42 at $T < 220$ K and $n_s$ values up to $1 \cdot 10^{11}\,\text{m}^{-2}$, which are similar in magnitude to those inferred for desert dust (Wagner et al., 2018).

Due to the intrinsic heterogeneous ice nucleation ability of purely inorganic, partly deliquesced sea salt particles below 220 K, the CFDC ice nucleation measurements with the aerosolised sea surface microlayer samples as well as algal and bacterial cultures were primarily conducted between 235 and 220 K in order to explore whether the additional biogenic organic constituents induced heterogeneous ice formation in the temperature range where the inorganic salt components are not yet ice-active entities. Ladino et al. (2016) probed the ice nucleation ability of the exudates of the three phytoplankton species *Thalassiosira pseudonana*, *Nannochloris atomus*, and *Emiliania huxleyi*, as well as the marine bacterium *Vibrio harveyi* at 233 K with the University of Toronto (UT)–CFDC. The exudates were separated from the cells with 0.1, 0.2, and 0.8 μm membrane filters to remove cell debris. The filtered solutions were aerosolised with an atomiser and dried, 500 nm-sized particles were selected for the CFDC measurements. The exudate particles only revealed a small heterogeneous freezing mode that was independent of the pore size of the filter, indicating that the ice-nucleating entities were smaller than 0.1 μm in size. The heterogeneous freezing onsets were very low ($S_{ice}$ ~1.05), but the ice-active particle fraction was only in the range between 0.001 and 0.01% until reaching $S_{ice}$ ~ 1.4, where the homogeneous freezing mode became apparent. Another phytoplankton species, the cyanobacterium *Prochlorococcus*, was probed on its ice nucleation ability with the Spectrometer for Ice Nuclei (SPIN)–CFDC (Wolf et al., 2019). Here, the seawater solution with the cell cultures (average concentration about $5 \cdot 10^8$ cells/mL) was aerosolised with a glass frit bubbler. Without prior treatment, the organic carbon content of the generated particles, as measured with an aerosol mass spectrometer, was very low and indicated that the 0.5 – 0.7 μm sized *Prochlorococcus* cells were not efficiently aerosolised. Concomitantly, ice formation by these particles at 227 K only initiated at the homogeneous freezing level. The organic carbon content and ice nucleation behaviour strongly changed when the cell cultures were treated by sonication prior to aerosolisation, causing the destruction of the outer cell membranes and releasing intracellular organic material (cell lysis). The increase in the organic carbon content after cell lysis was particularly pronounced for smaller particle sizes (200 nm). These 200 nm-sized particles proved to be very efficient INPs, with ice-active fractions >1% at $S_{ice}$ >1.18 and corresponding $n_s$ densities that were similar in magnitude to those of other common INPs like mineral and soil dust (Wolf et al., 2019).

Two further CFDC ice nucleation studies have investigated field-collected sea surface microlayer samples. Wilson et al. (2015) probed a variety of microlayer samples from the North Pacific and the British Columbia coastline. The samples were aerosolised with an atomiser, and the ice nucleation behaviour of dried, 200 nm-sized particles was investigated at 233 K with the UT-CFDC. The aerosol particles generated from the microlayer samples nucleated ice heterogeneously, with $S_{ice}$ onsets varying between 1.15 and 1.33 (representative of an ice-active fraction of 1%). It was found that the $n_s$ densities of the microlayer samples at 233 K and $S_{ice} = 1.35$ were typically around $1 \cdot 10^{11}$ m$^{-2}$, about one order of magnitude larger than data for kaolinite, feldspar, and Arizona test dust particles (Wilson et al., 2015). Most recently, Wolf et al. (2020) have collected sea surface microlayer samples in the Eastern Tropical North Pacific Ocean and the Florida Straits. The seawater samples were aerosolised with an atomiser and dried, 200 nm-sized particles were probed on their ice nucleation ability with the SPIN-CFDC. In agreement with the measurements by Wilson et al. (2015), the aerosol particles generated from these microlayer samples also proved to be effective INPs, with $S_{ice}$ nucleation onsets for a 1% activated fraction ranging between 1.14 and 1.38 at temperatures between 227 and 231 K.

**1.3 Motivation for a new set of ice nucleation measurements at cirrus conditions in the AIDA cloud chamber**

The available data allow the following preliminary conclusions regarding the ice nucleation behaviour of SSA particles and their proxies at cirrus temperatures. The heterogeneous ice nucleation experiments with the phytoplankton species mostly show a moderate reduction of the homogeneous freezing level to $S_{ice}$ onsets typically $\geq 1.3$. Notable exceptions are the very low onsets reported by Ladino et al. (2016) for the phytoplankton exudates and the high ice nucleation efficiency of the *Prochlorococcus* culture after sonication (Wolf et al., 2019). However, the very low onsets reported by Ladino et al. (2016) for the phytoplankton exudates were only representative for less than 0.01% of the particle ensemble. The particles generated from ambient sea surface microlayer samples by Wilson et al. (2015) and Wolf et al. (2020) typically nucleated ice well below the homogeneous freezing threshold, partly yielding an ice-active fraction of 1% already at a saturation ratio as low as $S_{ice} = 1.14$. They also revealed $n_s$ values that were larger than those of various types of dust particles. On the one hand, these results are reminiscent of recent laboratory experiments in the mixed-phase cloud temperature regime, showing that actively growing algal cultures do not sufficiently explain the ice nucleation activity inherent in some microlayer samples (Ickes et al., 2020). On the other hand, the high $n_s$ values of the particles generated from the microlayer samples in relation to dust at temperatures between 220 and 235 K are in contrast to results at mixed-phase cloud temperatures, where the $n_s$ values of SSA particles were reported to be typically two to three orders of magnitude lower than those of mineral and soil dust (DeMott et al., 2016). In this study we provide another data set of ice nucleation measurements with particles generated from field-collected sea surface microlayer and surface seawater samples at cirrus temperatures in order to investigate whether the Wilson et al. (2015) and Wolf et al. (2020) data are indeed a representative measure for the particles' ice nucleation ability or whether, similar to the lesson learned from the large number of studies at mixed-phase cloud temperatures, we need to consider a much higher diversity in the particles' ice nucleation behaviour. There are three new aspects in our study. Firstly, we have focussed on a different geographical region, using seawater samples from three different locations in the Arctic (Eastern Canadian

Arctic, Greenland Sea, and the glacial fjord Kongsfjorden in Svalbard). Secondly, we have probed another common marine phytoplankton species, the diatom *Skeletonema marinoi*, whose ice nucleation behaviour has not yet been investigated at cirrus conditions. Thirdly, we have used a different technique for the ice nucleation measurements by performing moderate expansion cooling experiments in the AIDA cloud chamber. At low temperatures, organic-rich SSA particles could be highly viscous,

and kinetic limitations of water diffusion into the particles could strongly affect the ice nucleation pathways (Berkemeier et al., 2014; Lienhard et al., 2015; Price et al., 2015; Fowler et al., 2020). During an AIDA expansion experiment, the particles have more time to adjust to changes in the RH compared to the short residence time in a CFDC (~ 10 s), and a tool of optical measurement techniques like infrared spectroscopy as well as light scattering and depolarisation measurements are available to analyse the particles' phase state during expansion cooling.

Our article is organised as follows. Section 2 describes the collection and preparation of the seawater samples and the diatom culture (Sect. 2.1) as well as the aerosol particle generation (Sect. 2.2) and the technical details of the ice nucleation measurements (Sect. 2.3). As the central part of our article, Sect. 3 summarises the results of the ice nucleation experiments. Before addressing in detail the experiments under cirrus conditions, we present a short summary of the freezing behaviour of the bulk solutions in the mixed-phase cloud temperature region as measured with a cold-stage instrument (Sect. 3.1). In doing

so, we can assess whether the range of freezing temperatures of our seawater samples is representative of that from previous studies in the mixed-phase cloud region. After aerosolisation of the bulk samples, we tested the particles' ice nucleation behaviour at two different cirrus temperatures in the AIDA chamber, namely at 229 and 217 K. At 229 K, the inorganic salt constituents of the seawater samples did not act as ice-nucleating entities and we could selectively probe the potential heterogeneous ice nucleation ability of the biogenic organic components (Sect. 3.2.1). At 217 K, the inorganic salt constituents

(only partially deliquesced) were efficient INPs, and we could investigate whether this behaviour was altered by the additional organic constituents in the seawater samples and the diatom culture (Sect. 3.2.2). In Sect. 4, we discuss our findings in the context of the previous measurements summarised in Sect. 1.2 and present suggestions for future investigations.

## 2 Experimental

### 2.1 Sample collection and preparation

We have probed seawater samples from three Arctic locations. The sea surface microlayer samples from the Eastern Canadian Arctic and the Greenland Sea were collected with the glass plate technique during NETCARE (Irish et al., 2019) and from a hydrophilic Teflon film on a rotating drum during ACCACIA (Wilson et al., 2015) field expeditions, respectively, and some of them were already used in previous AIDA ice nucleation measurements that focussed on the mixed-phase cloud temperature region (Ickes et al., 2020). The seawater samples from Kongsfjorden were collected during rough sea conditions with a Niskin

sampler placed horizontally on the water surface. Therefore, we use the term "surface seawater samples". These samples likely contained neuston and non-living material present in the surface microlayer, but this material will have been heavily diluted with subsurface water both due to the sampling technique and the weather conditions. A closer description of the Kongsfjorden

sampling site, the meteorological conditions during sampling, and the analysis of the aquatic chemistry and bacterial abundance of these samples is presented in Appendix A. For pertinent details regarding the NETCARE and ACCACIA samples we refer to the cited literature. All investigated samples are listed in Table 2. To ensure their unique identification, we labelled the NETCARE and ACCACIA samples with the abbreviations STN and SML, respectively, as used in the original publications. The acronym KFJ was used for the Kongsfjorden samples. The measurements presented in this article were conducted during September and October 2017.

In addition to the microlayer samples, we used a laboratory-grown culture of *Skeletonema marinoi*, a widespread marine diatom that often dominates spring blooms in coastal waters (Kooistra et al., 2008; Johansson et al., 2019; Stenow et al., 2020). Its cell wall morphology is similar to that in *Thalassiosira pseudonana*, both of which feature layers of polysaccharides, including glucan, in the outer part of their cell walls, known as exopolymeric substances (Hoagland et al., 1993). We have already probed the ice nucleation ability of *Skeletonema marinoi* (strain number CCAP 1077/5 from the Gothenburg University Marine Algal Culture Collection) at mixed-phase cloud temperatures in the AIDA chamber, and used the same strain to produce a new stock for the present measurements at cirrus conditions according to the procedure described by Ickes et al. (2020). We called this sample SM100 (Table 2), where the number refers to a nutrient saturation of 100% during algal growth (Ickes et al., 2020). The cell concentration, as measured with the cell counter TC20 (Bio-Rad), reached a density of $2.85 \cdot 10^6$ cells/mL. In addition to this concentrated SM100 culture, we prepared a diluted algal suspension (SM100_dil) with a dilution factor of 50, using a 3.5 wt% solution of a synthetic sea salt mixture (Sigma-Aldrich, product number S9883) in ultrapure water (GenPure Pro UV ultrapure water system, Thermo Scientific) as diluting agent. The cell concentration in SM100_dil is representative of an abundance peak observed during a strong spring bloom (Saravanan and Godhe, 2010).

## 2.2 Aerosol particle generation and characterisation

For the ice nucleation measurements in the AIDA chamber, the seawater samples and the SM100 cultures were thawed, homogenised by shaking, and aerosolised with an ultrasonic nebuliser (GA2400, SinapTec). After passing through a pair of silica gel diffusion dryers that reduced the ambient RH to less than 3%, the particles were injected into the cooled AIDA chamber. Four representative number size distributions of the dried particles are shown in Fig. 1. They result from the combination of the size spectra of a scanning mobility particle sizer (SMPS, TSI, mobility diameter range: 0.014 – 0.82 μm) and an aerodynamic particle spectrometer (APS, TSI, aerodynamic diameter range: 0.523 – 19.81 μm). The nebuliser produced uniform distributions with median particle diameters centred between 0.65 and 0.8 μm, as summarised in the last column of Table 2. We note that nebulisation is a convenient technique to produce sufficient particle numbers to fill the large volume of the AIDA chamber (84 m$^3$) in a reasonable amount of time. The typical injection period was 20 – 30 min, yielding a final aerosol particle number concentration, $N_{aer}$, of about 500 cm$^{-3}$, as measured with a condensation particle counter (CPC, TSI). However, nebulisation is not a process that mimics the natural sea spray production mechanism. Regarding the field-collected oceanic surface water samples, one drawback of this technique might be that aerosolisation of the well-mixed microlayer and

surface seawater suspensions in the nebuliser does not account for the formation of small film droplets from bursting bubble caps which can be particularly enriched in organic material (O'Dowd et al., 2004; Ault et al., 2013; Prather et al., 2013). Nonetheless, the experiments with the particles generated by nebulising the undiluted microlayer and surface seawater samples should yield a good estimate of the average ice nucleation ability of the organic entities contained in them. Regarding the SM100 culture, we have to consider two aspects. Firstly, we have strong indications from the measurements during our previous ice nucleation experiments at mixed-phase cloud temperatures that neither intact cells nor fragments of *Skeletonema marinoi* cells were efficiently transferred from the bulk to the particle phase by nebulisation (Ickes et al., 2020). Electron microscope images of particles that were generated by nebulising the SM100 culture did not show any evidence of such intact cells or cellular debris. Therefore, we expect that any observed heterogeneous ice nucleation activity of SM100 is primarily related to its exopolymeric secretions. Secondly, in Ickes et al. (2020) we compared the median freezing temperature of droplets pipetted directly from the SM100 culture to that of droplets pipetted from the condensate that was collected from the outflow of the ultrasonic nebuliser. Nebulisation increased the median freezing temperature of SM100 by about 2.5 K, presumably due to the vigorous mechanical action that facilitated the detachment of exopolymeric substances from the cell surface or led to the disruption of cells or cell agglomerates and enhanced the release of ice-active intracellular components into the solution. A similar explanation was given by Wolf et al. (2019) regarding the strong effect of sonication on the ice nucleation activity of *Prochlorococcus*. Therefore, we expect that the ice nucleation experiments with the particles from the nebulised SM100 culture were likely more representative of an aged phytoplankton bloom under stress conditions like phage- and viral infections, oxidative and osmotic stress, toxins, etc., i.e., processes that increase organic matter exudation during bloom breakdown (Mühlenbruch et al., 2018). In contrast, the freezing measurements with droplets pipetted directly from the SM100 culture represent actively grown diatoms under balanced, light- and nutrient-replete conditions.

### 2.3 Ice nucleation measurement techniques

#### 2.3.1 The cold-stage instrument INSEKT

In order to confirm that the sea surface microlayer and surface seawater samples as well as the SM100 culture contained ice-active entities for inducing heterogeneous freezing in the mixed-phase cloud temperature regime, we investigated the freezing behaviour of 50 µL aliquots pipetted from the bulk samples in the cold-stage instrument INSEKT (Ice Nucleation Spectrometer of the Karlsruhe Institute of Technology) (Schneider et al., 2020). INSEKT is a custom-made version of an apparatus developed at the Colorado State University, known in the literature as the "CSU Ice Spectrometer" (Hill et al., 2016). The aliquots were pipetted into two sterile 96-well polypropylene PCR trays. These trays were then fitted into two ethanol-cooled aluminium blocks, whose temperature was linearly reduced at a rate of 0.3 K/min until all aliquots were frozen. The basic measure of INSEKT was the freezing temperature of each individual aliquot. Freezing induced a brightness change of the suspensions in the wells which was detected by a camera mounted above the aluminium blocks. By dividing the temperature-dependent number of frozen aliquots through the total number of aliquots, the fraction frozen (*FF*) curve throughout the temperature ramp

was calculated. The *FF* curves were corrected for the freezing point depression caused by the salts to derive the hypothetical freezing temperature in pure water according to the procedure described by Wilson et al. (2015) and Irish et al. (2019). From the *FF* curves, we computed the cumulative concentration of INPs per sample volume, $n_{INP}(T)$, according to Eq. (1) (Vali, 1971):

$$n_{INP}(T) = -\frac{ln\big(1 - FF(T)\big)}{V_{aliquot}} \tag{1}$$

For a subset of samples, we diluted the suspensions by a factor of 10 and 100 with ultrapure water to extend the measured $n_{INP}(T)$ spectrum to lower freezing temperatures and higher cumulative INP concentrations. Thereby, we achieved a better overlap with other cold-stage measurements in which smaller aliquot volumes of 1 μL were used (Wilson et al., 2015; Irish et al. 2019). The *FF* curve of ultrapure water was additionally measured as a reference.

## 2.3.2 The aerosol and cloud chamber AIDA

The ice nucleation experiments at cirrus conditions were conducted in the aerosol and cloud chamber AIDA of the Karlsruhe Institute of Technology. The operation of the AIDA chamber has been described in detail in a large number of publications, but we want to specifically refer to its description in our study on the ice nucleation behaviour of purely inorganic sea salt particles at cirrus temperatures (Wagner et al., 2018), because the modus operandi in that work was essentially the same as in our current study. Briefly, the AIDA chamber is an 84 m³-sized aluminium chamber that is housed in an isolating, temperature-controlled box whose interior can be cooled to a minimum temperature of about 183 K. We determined the mean AIDA gas temperature by averaging the measurements from a vertical array of thermocouple sensors with an estimated uncertainty of ± 0.3 K. We quantified the relative humidity of the chamber air by measuring the intensity profile of a rotational-vibrational water vapour absorption line at 1.37 μm with a tuneable diode laser spectrometer (Fahey et al., 2014). This measurement was performed in situ and specifically yielded the interstitial water vapour concentration bare of any contribution from condensed or frozen water contained in aerosol and/or cloud particles. The measured water vapour partial pressure was divided by the saturation water vapour pressures over ice and supercooled liquid water to compute the respective saturation ratios $S_{ice}$ and $S_{liq}$ with an uncertainty of ± 5% (Murphy and Koop, 2005).

For the injection of the aerosol particles generated from the seawater samples and the SM100 culture, the AIDA chamber was conditioned to almost ice-saturated conditions ($S_{ice} \sim 1$) as controlled by a coating layer of ice on the inner chamber walls. To achieve ice-supersaturated conditions and induce the formation of ice crystals by either homogeneous or heterogeneous nucleation, we performed an expansion cooling experiment by reducing the chamber pressure from typically 1000 to 850 hPa. The number concentration of the nucleated ice crystals in the course of the expansion cooling run, $N_{ice}$, was inferred from the records of two optical particle counters (OPCs, type welas, Palas GmbH, size range 0.7 – 240 μm). The ice crystals rapidly grew to sizes larger than those of the seed aerosol particles, so that it was convenient to define an optical threshold size of

about 10 μm and to selectively classify and count all particles above that size as ice crystals. By dividing $N_{ice}$ through $N_{aer}$, we computed the ice-active fraction of the aerosol particle population, *FF*, and by further dividing *FF* through the average dry particle surface area, $A_{aer}$, as obtained from the size distribution measurements (Fig. 1), we calculated the ice nucleation active surface site density, $n_s$. The uncertainty of $n_s$ was estimated to ± 40%, using error propagation with individual uncertainties of ± 20% for $N_{ice}$ as well as ± 34% for $N_{aer} \cdot A_{aer}$ (Ullrich et al., 2017). As will be shown in Sect. 3.2, the *FF* was always so low that the linear approximation in computing $n_s$ was valid (Niemand et al., 2012). To provide an estimate of the lower detection limit with respect to *FF* and $n_s$ during the AIDA expansion cooling experiments, we consider the lower limit case that the OPCs would only detect one single ice crystal during a measurement period of 100 s. Based on the sampling flow rate and the cross-sectional area of the welas OPC sensors, this would correspond to an ice particle number concentration of about 0.005 cm$^{-3}$. With $N_{aer}$ typically being ~ 500 cm$^{-3}$ (Sect. 2.2), the *FF* would be about 0.001% ($1 \cdot 10^{-5}$). Given this and a typical value of 2 μm$^2$ for $A_{aer}$, the lower detection limit in terms of $n_s$ would be $0.5 \cdot 10^7$ m$^{-2}$.

Apart from the basic instrumentation described above, we used infrared extinction as well as light scattering and depolarisation measurements to probe the phase state of the added aerosol particles and to detect possible phase changes during expansion cooling. Infrared extinction spectra of the aerosol particles were recorded in situ from 6000 to 800 cm$^{-1}$ at 4 cm$^{-1}$ resolution with a Fourier transform infrared spectrometer (FTIR, model IFS66v, Bruker) that was coupled to a multiple reflection cell of 166.8 m path length (Wagner et al., 2006). With the so-called SIMONE instrument, we measured the light-scattering intensities of the aerosol and cloud particles in the AIDA chamber at 2° and 178° scattering angles (Schnaiter et al., 2012). At 178°, the scattered light intensity was detected polarisation-resolved, enabling the computation of the backscattering linear depolarisation ratio, $\delta$. The use and interplay of all employed measurement techniques will be outlined in greater detail in Sect. 3.2.

## 3 Results of the ice nucleation measurements

### 3.1 INSEKT cold-stage measurements at mixed-phase cloud temperatures

A summary of the freezing experiments with the cold-stage instrument INSEKT is shown in Fig. 2. Panel (a) shows the temperature-dependent *FF* curves for all investigated samples, corrected for the freezing point depression by the salts (see Sect. 2.3.1). The data are colour-coded with respect to the sampling location (magenta: STN samples from the Canadian Arctic, red: SML samples from the Greenland Sea, blue: KFJ samples from Kongsfjorden, green: SM100 culture, and grey: background measurement with ultrapure water). The median freezing temperatures, $T_{50}$, (corresponding to a *FF* of 50%) of the field samples encompassed a range between 265.7 K (STN2, the most ice-active sample) and 256.5 K (STN1, the least ice-active sample). The relative activity within the subset of STN samples, with $T_{50}$(STN2) > $T_{50}$(STN7) > $T_{50}$(STN1), is in good agreement with the results from the original droplet freezing measurements with these solutions presented in Irish et al. (2019). The same is true for the SML samples, where we confirmed the relative order of $T_{50}$(SML12.5) > $T_{50}$(SML6) > $T_{50}$(SML10) > $T_{50}$(SML13) from Wilson et al. (2015). In the next paragraph, we present a quantitative comparison of the data which takes

into account the different aliquot volumes of the respective measurements (50 μL in our study and 1 μL in Irish et al. (2019) and Wilson et al. (2015)). The five individual samples from Kongsfjorden show little variation in their $T_{50}$, ranging from 261.9

to 263.1 K. This may be partly explained by the strong dilution with subsurface waters, leading to a homogenisation of the KFJ surface seawater samples in comparison with the higher degree of variability observed for the SML and STN microlayer samples. The activity of the SM100 culture was less than any of the field samples, but its $T_{50}$ of 254.7 K was still almost 5 K higher than the blank measurement with ultrapure water. As already briefly discussed in Sect. 2.2, mechanical forces induced by nebulisation increased the $T_{50}$ of SM100 by 2.5 K in our previous study (Ickes et al., 2020). It is therefore conceivable that

the generally higher ice nucleation activity of the field samples in relation to the untreated SM100 culture could be due to aging processes that have led to the break-up of phytoplankton and to the dispersal of efficient ice-nucleating entities from intracellular material. What controls the large variability of $T_{50}$ within the field samples is still an outstanding question. Regarding the STN sampling location, a negative correlation between the freezing temperature and the salinity of the samples was found. Irish et al. (2019) argued that the lower salinity might be connected to sea ice melt and terrestrial run-off, i.e.,

processes that could increase the number of efficient ice-nucleating entities like microorganisms and their exudates in the seawater.

In Fig. 2b, we present the cumulative INP concentrations as a function of temperature for our samples ($n_{INP}(T)$, see Eq. (1)). For all samples except STN2, the $n_{INP}(T)$ traces were merged from at least two independent measurements, (i) with the undiluted samples, and (ii), with the samples diluted by a factor of 10 with ultrapure water to reduce the concentration of INPs

and shift the $n_{INP}(T)$ spectrum to lower freezing temperatures. Due to the similarity of the freezing behaviour within the set of the Kongsfjorden samples, we only included two exemplary $n_{INP}(T)$ curves for KFJ2 and KFJ5 in Fig. 2b. Here, we additionally performed measurements with the solutions diluted by a factor of 100. For comparison, coloured horizontal bars indicate the range of temperatures where the individual microlayer samples from the collections probed by Irish et al. (2017) (black), Irish et al. (2019) (green), and Wilson et al. (2015) (orange) showed an INP concentration of $10^5$ and $10^6$ L$^{-1}$. The comparison

indicates that the $n_{INP}(T)$ curves from our new INSEKT measurements nicely fall into to the range of previous droplet freezing measurements that comprised a large variety of Arctic microlayer samples. For the STN and SML samples which have been re-measured with INSEKT after long-term storage of the frozen solutions at 193 K, we did not detect any pronounced alterations of the $n_{INP}(T)$ spectrum with respect to the original measurements. For example, the re-measured $n_{INP}(T)$ curve of STN2, our most active microlayer sample, was shifted by only about 1.5 K to lower temperatures in comparison with the

original measurement (Irish et al., 2019). This deviation can also be explained by the combined uncertainties of the two measurements and differences in the applied procedures e.g. with respect to the cooling rate. In spite of their dilution with subsurface water, the KFJ samples still contain a substantial INP concentration at high freezing temperatures above 260 K. However, the slope of the $n_{INP}(T)$ lines for the KFJ surface seawater samples tends to be flatter compared to that of the STN and SML microlayer samples, which might be due to the dilution effect. In summary, the INSEKT cold-stage measurements

suggest that our samples in their entirety are a representative cross-section of the heterogeneous ice nucleation ability typically

shown by Arctic microlayer suspensions at mixed-phase cloud temperatures. In the following section, we investigate the ice nucleation ability of aerosol particles generated from the bulk solutions at cirrus conditions in the AIDA chamber.

## 3.2 AIDA cloud chamber measurements at cirrus conditions

### 3.2.1 Start temperature 229 K, where inorganic salts do not contribute to heterogeneous ice formation

We first focus on the expansion cooling experiments started at about 229 K. Similar to the droplet freezing experiments in the mixed-phase cloud temperature regime, any observable heterogeneous ice nucleation mode must be related to organic material or other ice-nucleating entities like dust contained in the aerosol particles, because the inorganic salt components are not yet ice-active at this temperature. In Fig. 3, we provide an overview of the ice nucleation ability of our samples by showing the AIDA records from six individual expansion cooling runs. For each experiment, the time series of the AIDA data are divided

into four different panels; I: AIDA mean gas temperature (red line) and pressure (black line); II: saturation ratios with respect to ice ($S_{ice}$, brown line) and supercooled water ($S_{liq}$, magenta line); III: size-resolved single-particle scattering signals from the optical particle counter (green dots); IV: forward scattering intensity ($I_{for}$, 2° scattering angle, grey line) and backscattering linear depolarisation ratio ($\delta$, 178° scattering angle, blue line) from the SIMONE measurements. The start of pumping is set to time zero. Vertical dashed lines indicate prominent events during the expansion runs like particle deliquescence as well as

heterogeneous and homogeneous ice nucleation onsets. Green markings in panels III of Figs. 3b, d, and f direct the eye to small heterogeneous ice nucleation modes. Additionally, Fig. 4 shows a series of FTIR spectra that were recorded in the initial 80 seconds of the expansion run with the particles generated from the SM100 sample (one spectrum every 10 seconds), as indicated by the horizontal blue bar in panel IV of Fig. 3b. This time period covers the full deliquescence step of the internally mixed solid-liquid particles (see discussion below). The behaviour evident in Fig. 4 is representative of all conducted

experiments and, therefore, the FTIR spectra are only displayed once.

Figure 3a shows the previously measured ice nucleation behaviour of particles generated from a commercially available bulk Atlantic water sample (data from Fig. 6, upper left panel, of Wagner et al. (2018)). We consider this measurement as the blank experiment for the intrinsic ice nucleation behaviour of purely inorganic SSA particles when probed in an expansion cooling run started at ~ 229 K. The same behaviour as displayed in Fig. 3a was also observed when using a solution of the synthetic

Sigma-Aldrich sea salt mixture for particle generation (Wagner et al., 2018). It is important to note that we have mimicked the procedure of the previous CFDC ice nucleation measurements with sea surface microlayer samples and have dried the aerosol particles from the outlet of the nebuliser before injecting them into the AIDA chamber. The hygroscopic behaviour and the phase change of the particles upon injection and during the initial time period of the expansion run are revealed by the SIMONE measurements (Fig. 3a, panel IV) and the representative FTIR records shown in Fig. 4. In a similar experiment with dried NaCl

particles of the same size, the depolarization ratio $\delta$ of the added crystals was about 25% and the concomitant FTIR spectrum did not show any signature of liquid water (Wagner et al., 2018). In contrast, after injecting the dried particles from the bulk Atlantic water sample into the AIDA chamber at 229 K and nearly ice-saturated conditions, $\delta$ was only about 4% and the

characteristic liquid water absorption bands were clearly visible in the infrared extinction measurement, e.g. in the wavenumber region between 3500 and 2800 cm$^{-1}$ related to the O–H stretching mode (lowermost spectrum in Fig. 4 recorded at $t = 0$). In

agreement with the results of Schill and Tolbert (2014), the particles were thus in an internally mixed solid-liquid state, with a brine layer of dissolved salts surrounding the solid NaCl core. Even if having a spherical outline, such particles are still inhomogeneous from the viewpoint of light scattering and caused the small residual depolarisation of 4% (Sun et al., 2011). During the first 40 seconds of the expansion run, the liquid layer of the particles slightly grew through the uptake of water vapour from the gas phase as a result of the increasing relative humidity. This led to a small reduction of $\delta$, a slight increase

of $I_{\mathrm{for}}$ due to particle growth by the water uptake, and a small increase of the intensity of the liquid water absorption bands in the FTIR spectra. At about 40 s (first vertical line in Fig. 3a), these smooth variations turned into a sudden, stepwise increase of $I_{\mathrm{for}}$ and, simultaneously, a strong growth of the liquid water absorption band intensities in the FTIR spectra was detectable. Also the scattering contribution to the infrared extinction spectra at wavenumbers > 4000 cm$^{-1}$ strongly increased due to the growth of the particles caused by water uptake. At these conditions ($t = 40$ s, $S_{\mathrm{ice}} = 1.13$, $S_{\mathrm{liq}} = 0.73$), we had reached the full

deliquescence point of the sea salt particles (Koop et al., 2000a). The internally mixed solid-liquid particles underwent a phase transition to homogeneously mixed aqueous solution droplets, for which the depolarisation ratio finally dropped to the background value of zero. Ice nucleation by the solution droplets in the further course of the expansion run occurred at the homogeneous freezing level ($S_{\mathrm{ice}} = 1.48$, second vertical line in Fig. 3b). The formation of ice crystals was indicated by the appearance of the mode of big particles in the OPC records at sizes larger than 10 μm, the further strong increase of $I_{\mathrm{for}}$, and

the renewed increase of the depolarisation ratio due to the presence of aspherical ice particles.

How did the ice nucleation behaviour change when the sea salt aerosol particles contained additional organic components? In Fig. 3b, we show the AIDA records from the expansion run with the particles generated from the undiluted SM100 culture with a concentration of $2.85 \cdot 10^6$ cells/mL. Just at the onset of the full deliquescence step ($S_{\mathrm{ice}} \sim 1.1$, first vertical line), we detected a very small heterogeneous ice nucleation mode with an activated fraction, $FF$, of 0.003%. The few nucleated ice

crystals are highlighted by the green circle in panel III. A much stronger immersion freezing mode was observed in closer proximity to the homogeneous freezing threshold, starting at $S_{\mathrm{ice}} = 1.38$ (second vertical line). Here, a maximum activated fraction of 0.7% was observed before the onset of the homogeneous freezing mode at $S_{\mathrm{ice}} \geq 1.48$ led to a further increase of the ice particle number concentration (third vertical line). The two independent ice nucleation modes that started at $S_{\mathrm{ice}} = 1.38$ (immersion freezing) and 1.48 (homogeneous freezing) can clearly be seen by the two stepwise changes in the time series of

$I_{\mathrm{for}}$ and $\delta$ (panel IV). Diluting the SM100 culture by a factor of 50 strongly decreased the magnitude of the heterogeneous freezing modes, with the first mode at $S_{\mathrm{ice}} = 1.1$ completely disappearing and the second mode at $S_{\mathrm{ice}} = 1.38$ reduced to a $FF$ of 0.04% (Fig. 3c).

The AIDA records of three exemplary expansion cooling runs with particles generated from the microlayer and surface seawater samples STN2, SML13, and KFJ4 support the finding from the experiment with SM100 that there are two distinct

regimes where a heterogeneous ice nucleation activity becomes apparent (Figs. 3d, e, and f). The particles generated from the

STN2 sample showed ice formation in both regimes, i.e., the early, weak immersion freezing mode starting at $S_{ice} = 1.1$ (first vertical line in Fig. 3d, nucleated ice crystals again highlighted by the green marking in panel III) as well as the later, more pronounced nucleation mode in the range $S_{ice} = 1.38 - 1.48$ (i.e., between the second and third vertical line in Fig. 3d). For SML13 and KFJ4, however, we only detected heterogeneous ice formation in one of the two ranges, i.e., for SML13 at $S_{ice}$ between 1.38 and 1.48, and for KFJ4 at $S_{ice} = 1.1$. The fact that the two regimes are not interrelated suggests that different types of ice-nucleating entities are responsible for the observed nucleation modes. To quantify the heterogeneous ice nucleation ability of the particles, we have therefore divided the range of ice saturation ratios into two regions, (i), $S_{ice} = 1.10 - 1.38$, and (ii), $S_{ice} = 1.38 - 1.48$, and have summarised in Table 3 the maximum values of $FF$ and $n_s$ that occurred in these two ranges. In the first regime, the $FF$ data were extremely low with a maximum value of only 0.01%, and the corresponding ice nucleation active surface site densities were on the order of $10^7$ m$^{-2}$, very close to the estimated detection limit of our OPC measurements (see Sect. 2.3.2). Nonetheless, we are confident that these small nucleation modes are related to the particles generated from the SM100 culture and the surface seawater samples and are not caused by the chamber background for two reasons: Firstly, for the simple reason that some experiments did not show this early nucleation mode at all, and secondly, because repeated reference expansion runs with the background aerosol particles present in the cleaned AIDA chamber (typically ~ 0.1 cm$^{-3}$) only led to ice formation at or above the homogeneous freezing level. In the second heterogeneous ice nucleation regime at $S_{ice}$ between 1.38 and 1.48, the maximum $FF$ and $n_s$ values were typically one order of magnitude larger than in the first regime, with the two most ice-active samples SM100 and SML13 even exceeding an ice nucleation active surface site density of $10^9$ m$^{-2}$. The $n_s$ value for the diluted *Skeletonema marinoi* culture, SM100_dil, was of the same order as for most of the field samples. As a comparison, the $n_s$ values from Table 3 are about two orders of magnitude smaller than those derived from AIDA expansion cooling runs with desert dust particles at cirrus temperatures (Ullrich et al., 2017).

An intriguing question is whether one can relate the heterogeneous ice nucleation ability of the particles at cirrus conditions to the freezing behaviour of the bulk solutions at mixed-phase cloud temperatures. To facilitate such comparison, we have included in the last column of Table 3 the $T_{50}$ freezing temperatures of the bulk samples from the data shown in Fig. 2a. In both temperature regimes, the underlying ice formation pathway is immersion freezing, but as discussed in Sect. 2.2, aerosolisation might affect the amount and identity of the ice-nucleating entities, in particular in the experiment with the SM100 culture. Nonetheless, we can draw some tentative conclusions. One noticeable tendency is that the respective bulk solutions of particles that revealed a comparably strong nucleation mode at low saturation ratios ($S_{ice} = 1.10 - 1.38$) in the AIDA expansion runs like STN2, SML6, KFJ4, and KFJ5 also showed a relatively high $T_{50}$ freezing temperature $\geq 262$ K in the INSEKT measurements. One might therefore speculate that the few, but very efficient ice-nucleating entities which caused the very small ice nucleation mode at low supersaturations in the AIDA experiments could be the same that were responsible for the freezing of the sample aliquots at high temperatures ($\geq 262$ K). However, due to the extremely low $FF$ values from the AIDA experiments and the correspondingly large uncertainties, we refrain from an in-depth statistical correlation analysis between $FF$ and $T_{50}$. A somewhat more secure conclusion is that the occurrence of a pronounced heterogeneous ice nucleation

mode at higher ice saturation ratios in the AIDA chamber ($S_{ice} = 1.38 – 1.48$) is not linked to a particularly high freezing temperature of the respective bulk solutions. As mentioned above, the particles generated from the SML13 and SM100 samples showed the highest $n_s$ values in that $S_{ice}$ range, but the corresponding $T_{50}$ freezing temperatures of the bulk solutions belonged to the lower end of the spectrum of observed freezing temperatures. We will resume this discussion in Sect. 4 when we compare our new AIDA results with the previous ice nucleation measurements summarised in the introduction.

### 3.2.2 Start temperature 217 K, where inorganic salts contribute to heterogeneous ice formation

Figure 5a shows the AIDA records of an experiment where the ice nucleation ability of the particles generated from the bulk Atlantic water sample was probed at a lower starting temperature of 217 K (data from Fig. 6, upper right panel, of Wagner et al. (2018)). The key difference compared to the data shown in Fig. 3a is that we see the initial water uptake by the brine layer (small reduction in $\delta$, small increase in $I_{for}$), but the full deliquescence step does not occur and instead a dominant nucleation mode of ice crystals starts to form at $S_{ice} = 1.24$ due to immersion freezing by the still undissolved particle core primarily composed of NaCl (vertical line in Fig. 5a). The activated fraction is so large ($FF \sim 10\%$) that the nucleated ice crystals rapidly deplete the excess of water vapour, limiting the peak saturation ratio during the expansion run to a maximum value, $S_{ice,max}$, of 1.30. In this case the relevant question is: Will the additional organic compounds in the microlayer samples and the SM100 culture have any influence on this strong heterogeneous ice nucleation ability already shown by the inorganic salt components? Two exemplary AIDA data sets from expansion runs with particles generated from the SM100 and KFJ1 samples are shown in Fig. 5b and c. Through visual inspection alone, the time series of the saturation ratios (panel II) and the scatter plots with the OPC data (panel III) look unchanged in comparison with the blank experiment shown in Fig. 5a. A quantitative analysis is provided in Table 4. Here, we have tabulated the $S_{ice}$ nucleation onsets for a $FF$ of 0.1% and have computed the $n_s$ values at the peak ice saturation ratio that was reached during the individual expansion runs. The data underline the very high similarity of the ice nucleation behaviour between the various samples and, in particular, that there is no notable difference between the blank experiment with the bulk Atlantic water sample and the experiments with the SM100 culture or the microlayer and surface seawater samples. All $S_{ice}$ onsets and $n_s$ values fall into the compact ranges of $1.24 – 1.28$ and $4.9 – 6.1 \cdot 10^{10}$ m$^{-2}$, respectively. Note that these $n_s$ values, which represent the immersion freezing mode of the inorganic salt components at 217 K, are typically two to three orders of magnitude larger than those representing the immersion freezing mode of the organic compounds contained in the microlayer and surface seawater samples as well as the SM100 culture at 229 K (Table 3).

As a summary of our observations, we show in Fig. 6 two diagrams in the $S_{ice}$ versus $T$ space to illustrate the difference in the ice nucleation abilities at 229 and 217 K, contrasting the behaviour of the particles from the bulk Atlantic water sample (blank, Fig. 6a) with that of the particles generated from the microlayer sample STN2 (Fig. 6b). The grey lines show the trajectories of the AIDA expansion experiments started at 229 and 217 K, covering the period from the start of pumping to the time when the maximum $S_{ice}$ value was reached. The size of the purple-coloured circles superimposed on the trajectories denotes the $FF$ encountered in the course of expansion cooling. Reference lines indicate the homogeneous freezing onset of aqueous solution

droplets (blue line, Koop et al. (2000b)), the saturation curve with respect to supercooled liquid water (red line), and ice-saturated conditions (dashed black line). The estimated RH range of the full deliquescence (FDRH) of inorganic sea salt particles is indicated by the orange shaded area. At 298 K, Tang et al. (1997) observed the full dissolution of levitated sea salt aerosol particles at RH between 71 and 74% RH, whereas pure NaCl particles deliquesced at 75.3% RH. We therefore scaled

the extrapolated, temperature-dependent parameterisation of the deliquescence relative humidities of pure NaCl particles from Tang and Munkelwitz (1993) with an absolute, temperature-independent shift between –4% and –1% on the relative humidity scale to estimate the RH range for the full dissolution of the inorganic sea salt particles at low temperatures. The measured onsets for the full deliquescence in the expansion runs started at 229 K are indicated by the blue stars. Whereas the fully deliquesced particles from the bulk Atlantic water sample only started to nucleate ice at the homogeneous freezing threshold

when probed at 229 K, the particles from the STN2 microlayer sample exhibited two weak heterogeneous ice nucleation modes prior to homogeneous freezing, with the first mode occurring just at the onset of the full deliquescence step ($FF \sim 0.01\%$), and the second mode at $S_{ice} \sim 1.38$ ($FF \sim 0.1\%$). Much higher $FF$ values in the order of 10% were observed for both particle types in the expansion runs started at 217 K. Heterogeneous ice formation started just before the predicted FDRH of the particles was reached, and the rapid increase of the $FF$ after the nucleation onset limited $S_{ice}$ to peak values of 1.30 – 1.35. The green

box and the green bar in Fig. 6b (with labels I and II) indicate the range of onset conditions for ice nucleation observed in the CFDC experiments by Wolf et al. (2020) and Wilson et al. (2015). Section 4.2 will address the comparison of this literature data with our results.

## 4 Discussion and outlook

### 4.1 Ice nucleation experiments with phytoplankton species

Our new AIDA results on the ice nucleation behaviour of particles generated from microlayer and surface seawater suspensions and diatom cultures at cirrus temperatures show both similarities and discrepancies with the data from the previous studies that we have summarised in Sect. 1.2. In Fig. 7, we present a compilation of the freezing data for the experiments with phytoplankton and marine bacterial cells as well as their exudates. The grey line shows the trajectory of our AIDA expansion experiment with the particles generated from the SM100 culture in the $S_{ice}$ versus $T$ space. The light green coloured diamonds

denote the two heterogeneous freezing onsets determined during the expansion run, with the first one at $S_{ice} = 1.1$ representing the very small freezing mode with $FF = 0.003\%$ and the second one at $S_{ice} = 1.38$ the dominant nucleation mode with $FF = 0.7\%$ (Fig. 3b, first and second vertical line, respectively). The light blue coloured diamond shows the onset of homogeneous freezing (third vertical line in Fig. 3b). The onset of the dominant heterogeneous ice nucleation mode of *Skeletonema marinoi* (SM100) is in very good agreement with the freezing data for cells of *Thalassiosira pseudonana* and *Nannochloris atomus* in

aqueous NaCl solution droplets (Alpert et al., 2011a, b) as well as the exudate freezing onsets of *Thalassiosira pseudonana* (Wilson et al., 2015). These algal species show a similar reduction of the $S_{ice}$ nucleation onsets by about 0.10 – 0.15 with respect to the homogeneous freezing line, whereas the freezing data for the coccolithophore *Emiliania huxleyi* coincide with

those measured for pure NaCl solution droplets (Alpert et al., 2011a). It is notable that the heterogeneous freezing mode is not only linked to the intact diatom cells, but also to the exudate material present in the diatom cultures. This follows both from the results of Wilson et al. (2015) for exudates of *Thalassiosira pseudonana* and from our experiments with the *Skeletonema marinoi* culture, where we argued that nebulisation of the cell suspension did not lead to the transfer of intact cells to the aerosol phase. Given this, ice-active material released by phytoplankton exudation might also be responsible for the heterogeneous freezing mode at $S_{ice}$ between 1.38 and 1.48, which we have observed in several experiments with the particles generated from the microlayer samples.

Ladino et al. (2016) detected in their CFDC measurements a very small early ice nucleation mode at $S_{ice} \sim 1.05$ for the exudates of *Thalassiosira pseudonana*, *Nannochloris atomus*, *Emiliania huxleyi*, and *Vibrio harveyi* with *FF* values between 0.001 and 0.01% (green pentagon in Fig. 7). This is also in agreement with our measurements, where we observed an equally small early nucleation mode for SM100 at $S_{ice} = 1.1$ with *FF* = 0.003%. Since this nucleation mode was enhanced in some of the surface seawater samples, we speculate that it is only partially attributable to ice-active material from fresh phytoplankton cells and may also be related to various altering pathways that produce additional or different types of ice-nucleating entities. Such pathways may include, on the one hand, the processing of exudates either through biological processes such as microbial metabolism (Wang et al., 2015; McCluskey et al., 2017) or physicochemical processes such as photochemistry and, on the other hand, mechanisms such as cell lysis by which additional intracellular organic material can be released. The latter effect was clearly shown by the ice nucleation experiments with *Prochlorococcus* (Wolf et al., 2019). Here, particles generated from an untreated cell suspension nucleated ice at the homogeneous freezing threshold while cell lysis induced by sonication reduced the particles' freezing onset to $S_{ice} \sim 1.18$ (see Fig. 5 in Wolf et al., 2019). It should be emphasised that this nucleation onset already corresponded to a *FF* of 1%, underlining the strong effect on the ice nucleation behaviour that was induced by the targeted treatment of the *Prochlorococcus* cells by sonication. Similarly, the heterogeneous freezing mode of SM100 at $S_{ice} = 1.1$ could also be related to the additional release of ice-active intracellular material when using the ultrasonic nebuliser for aerosol particle generation. However, the amount of dispersed ice-nucleating entities was obviously much smaller than in the Wolf et al. (2019) study. This could be due to the different cell concentrations of the suspensions examined in our work compared to those in Wolf et al. (2019), i.e., $2.85 \cdot 10^6$ cells/mL of *Skeletonema marinoi* vs. $5 \cdot 10^8$ cells/mL of *Prochlorococcus*. Additional ice nucleation experiments performed by Wolf et al. (2019) with a variety of organic compounds indicated that certain proteins and carbohydrates like aspartic acid, amylopectin, and agarose could be responsible for the improved ice nucleation ability of the particles from the lysed *Prochlorococcus* cultures. Such additional ice-nucleating entities might also explain the particularly high freezing temperature of some microlayer samples ($T_{50} \geq 262$ K) in the bulk freezing measurements with INSEKT, whereas aliquots of the pristine SM100 culture only froze at a $T_{50}$ of 254.7 K. The latter value is in reasonable agreement with the freezing temperature of 250 K reported for *Thalassiosira pseudonana* cells in pure water droplets (Knopf et al., 2011).

## 4.2 Ice nucleation experiments with sea surface microlayer and surface seawater samples

Whereas our ice nucleation experiments with SM100 fit well into previous data, the same is not true for the ice nucleation experiments with our microlayer and surface seawater samples. Both the particles from the samples collected by Wilson et al. (2015) and by Wolf et al. (2020) showed much higher ice-active fractions at temperatures above 220 K. As noted above, the range of onset conditions for exceeding a $FF$ of 1% in those measurements is depicted by the green bar and the green box in Fig. 6b. The critical saturation ratios to exceed a $FF$ of 1% were as low as $S_{ice} = 1.15$ ($T = 233$ K) for the particles from the Pacific microlayer samples investigated by Wilson et al. (2015), and the inferred $n_s$ values reached a magnitude of about $1 \cdot 10^{11}$ m$^{-2}$ at $S_{ice} = 1.35$. Wolf et al. (2020) observed a strong reduction of the nucleation onsets ($FF = 1\%$) for the particles from the Eastern Tropical North Pacific Ocean and the Florida Straits from $S_{ice} \sim 1.3$ to $S_{ice} \sim 1.15$ when reducing the temperature from 231 to 227 K. For the North Pacific sampling region, even the particles from subsurface seawater samples collected between 2 and 5 meters below the surface showed a dominant heterogeneous freezing mode with an average $S_{ice}$ onset for $FF = 1\%$ of about 1.25 at 227 K (Wolf et al., 2020). In contrast, we did not even observe a heterogeneous freezing mode with $FF \geq 1\%$ for the particles from our Arctic microlayer and surface seawater samples in the expansion runs started at 229 K. Amongst our samples, SML13 showed the highest ice-active fraction, amounting to $FF = 0.22\%$ in the nucleation regime between $S_{ice} = 1.38$ and 1.48, corresponding to a $n_s$ value of $1.1 \cdot 10^9$ m$^{-2}$, i.e., two orders of magnitude lower than typically observed for the Pacific samples probed by Wilson et al. (2015).

Due to the limited number of measurements, it is premature to ascribe this difference solely to the geographical sampling region, i.e., the Arctic region in our study versus locations in temperate and subtropical zones in the studies by Wilson et al. (2015) and Wolf et al. (2020). A factor that could contribute to a regional variation in the INP concentrations is the biogeographic pattern of the phytoplankton species. As summarised in Sect. 1.2, there are notable variations in the heterogeneous ice nucleation ability of various phytoplankton species under cirrus conditions, with e.g. *Prochlorococcus* showing distinctly lower critical ice saturation ratios compared to *Thalassiosira pseudonana*, *Nannochloris atomus*, and *Emiliania huxleyi*. The phytoplankton species richness in the tropics was found to be about three times that in higher latitudes (Righetti et al., 2019), potentially increasing the probability that a particularly ice-active species can be found in field-collected microlayer samples from tropical regions. *Melosira arctica*, the most productive algae in the Arctic Ocean (Booth and Horner, 1997), was not a source of particularly active INPs in our previous AIDA ice nucleation measurements that focussed on the mixed-phase cloud temperature region (Ickes et al., 2020). Differences in the local biological activity could influence the organic carbon enrichment in the SSA particles, and thereby affect their ice nucleation ability (Wolf et al., 2020). However, it is still unclear to what extent the organic enrichment in sea spray aerosol is controlled by the primary productivity in marine environments, which is characterised by chlorophyll-a levels of seawater as a measure of phytoplankton biomass. Several studies have found that the organic matter enrichment in sea spray is directly linked to primary production (e.g. Ceburnis et al., 2011, 2016; van Pinxteren et al., 2017). Other studies have reported that the size-resolved organic mass fractions were relatively invariant for a wide range of phytoplankton biomass and a broad diversity of phytoplankton components (e.g. Quinn

et al., 2014; Bates et al., 2020). Quinn et al. (2014) therefore concluded that local biological activity is of minor importance and uncoupled from a large reservoir of organic carbon in ocean surface waters, which primarily controls the enrichment of organic matter in SSA particles.

When comparing different measurements, it is important to take into account that there can also be a strong seasonal variation in the INP concentrations for the same sampling location and that the measured ice nucleation ability might also depend on the thickness of the sampled microlayer, i.e., how much ice-active organic material was sampled in relation to inorganic solutes (Irish et al., 2017; Irish et al., 2019), and therefore on the technique used to acquire the sample. Nonetheless, it is remarkable that the heterogeneous ice nucleation activity of all our samples is consistently very low at cirrus conditions, provided that the inorganic salts are not yet ice active. The bulk freezing measurements with INSEKT clearly indicated that our samples contained representative amounts of ice-nucleating entities at mixed-phase cloud conditions. A quantitative comparison between ice nucleation under mixed-phase cloud and cirrus conditions is challenging because different nucleation modes might be involved. As noted in the introduction, the $n_s$ values of SSA particles are typically two to three orders of magnitude lower than those of mineral and soil dust under mixed-phase cloud conditions (DeMott et al., 2016). For the particles from our microlayer and surface seawater samples, we observed the same poor heterogeneous ice nucleation ability in comparison with mineral and soil dust also under cirrus conditions down to a temperature of about 220 K, below which the crystalline inorganic salts became ice-active.

### 4.3 Influence of the aerosolisation method

A key factor in all laboratory experiments with microlayer samples and phytoplankton species is the aerosolisation method. Both in Wilson et al. (2015), Wolf et al. (2020), as well as in our study, the particles were produced from well-mixed microlayer and surface seawater samples with standard aerosol generators, so that we do not have any reason to assume that the observed differences in the ice nucleation activity are related to the aerosol generation method. However, all of these measurements only represent some kind of averaged ice nucleation activity, meaning that the ice-nucleating entities are equally distributed amongst all particles and that there is presumably no significant variability in the particle composition. As already mentioned in Sect. 2.2, these techniques do not mimic the natural process of sea spray aerosol production, where the bursting of bubble cap films can lead to the formation of highly organically enriched particles (O'Dowd et al., 2004; Ault et al., 2013; Prather et al., 2013). For ice nucleation experiments under cirrus conditions, the particle composition is particularly important because it can influence the underlying ice nucleation mode. For particles predominantly or even exclusively composed of organics, the ice nucleation mode might change from immersion freezing, as observed in the AIDA experiments, to deposition nucleation, where ice formation initiates by the deposition of water vapour on crystalline or glassy surfaces (Murray et al., 2010; Wilson et al., 2012). For this reason, Wolf et al. (2019) have used pure organic compounds as a proxy to represent the ice nucleation ability of the particles from the lysed *Prochlorococcus* cells.

In our previous ice nucleation experiments at mixed-phase cloud temperatures, we have attempted a more representative way of SSA production and have added 80 to 900 mL volumes of microlayer suspensions and diatom cultures to 20 L of artificial seawater in a sea spray simulation chamber (Ickes et al., 2020). Here the entrainment of air and associated bubble formation was induced by a plunging jet of water (Christiansen et al., 2019). The major drawback of this method was the extremely long injection period of 14 – 16 h to fill the 84 m³ volume of the AIDA chamber with a sufficiently high particle number concentration of 300 – 400 cm⁻³. In our current study, we have used the nebuliser instead of this time-consuming technique because we wanted to get an overview of the ice nucleation activities of many individual samples and directly compare them to those found by Wilson et al. (2015) and Wolf et al. (2020), i.e., studies where the particles were also generated from well-mixed microlayer samples. However, we consider the additional use of such sea spray simulation chambers or wave channels for mimicking SSA generation as one of the two most important directions for future laboratory studies on the SSA particles' ice nucleation ability at cirrus conditions. The associated questions would be twofold: Even if the average ice nucleation activity of the particles from a specific microlayer sample were small (as shown in the AIDA experiments), could it be possible that a larger proportion of ice-active particles is formed when the sample is added to the sea spray simulation chamber, with the chance that purely organic or highly organic-rich particles are also formed? And the question formulated in the opposite direction would be: Even if the average ice nucleation activity of the particles from a well-mixed microlayer sample were high (like in the samples probed by Wilson et al. (2015) and Wolf et al. (2020)), how much of this activity would be retained if we added the microlayer sample to the sea spray tank and mimicked a more natural way of particle production? In our experiments at mixed-phase cloud conditions, we observed a very diverse behaviour, with some microlayer samples retaining their ice nucleation activity when added to the sea spray tank, while for others the activity was significantly reduced (Ickes et al., 2020). As previously mentioned, under cirrus conditions the outcome of such experiments is additionally complicated by the potential change in the ice nucleation mode depending on whether particles with a high or with a low amount of organic substances are formed.

## 4.4 Influence of the ice nucleation measurement technique

The second one of the two most important factors in future studies would be the intercomparison of different ice nucleation measurement techniques. The most recent laboratory workshop on the intercomparison of ice nucleation measurements has focussed on immersion freezing experiments under mixed-phase cloud conditions (DeMott et al., 2018). We consider it equally important to perform such a study under cirrus conditions where, due to their complex hygroscopic behaviour, SSA particles would be an interesting and experimentally challenging INP type to be investigated. Let us imagine an aerosol particle that was produced from a microlayer sample, dried to a low relative humidity, and is now in a supersaturated environment with $S_{ice} = 1.3$ at 225 K, on the one hand suspended in the AIDA chamber and on the other in transit through the flow region of a CFDC. In the AIDA chamber, the particle has undergone a long RH history before reaching the specified conditions. It was first added to the chamber at $S_{ice} \sim 1$ and kept there for at least 20 min during the size distribution measurements. Then, it was subjected to a moderate expansion cooling cycle and in the course of about 100 s the ice saturation ratio has increased to 1.3.

The AIDA measurements with the Arctic surface seawater samples clearly revealed the various deliquescence steps the particle had time to undergo, i.e., the partial deliquescence already upon injection at ice-saturated conditions and the full deliquescence step in the initial period of the expansion run. Although the ice saturation ratio in the flow region of a CFDC can also be increased smoothly, the instrument is constantly flushed with fresh aerosol particles. The aerosol is surrounded by two sheath air flows, so that the sample temperature and the water vapour environment are very narrow and well defined (Rogers, 1988). The location of the aerosol lamina and its associated temperature and supersaturation conditions can be accurately calculated from instrumental parameters such as wall temperatures, sheath flow rates, and sample flow rates (Rogers, 1988; Kulkarni and Kok, 2012; Garimella et al., 2016). Computational fluid dynamics calculations show that the initially warm and dry sample air flow quickly adopts the nominal lamina temperature and $S_{ice}$ value within the upper 5–10% section of the main chamber (Garimella et al., 2016). As such, an aerosol particle in the centre of the flow region of a CFDC at $S_{ice}$ 1.3 and 225 K is instantly subjected to these ice supersaturated conditions after drying and has not experienced the same RH history as in the AIDA chamber. Its transit time through the nucleation region of a CFDC is typically about 10 s (Rogers, 1988). Recent measurements with the SPIN-CFDC have shown that the instrument is capable of detecting the deliquescence of inorganic sea salt particles even at low temperature, meaning that the deliquescence occurs at least within a couple of seconds (Kong et al., 2018). However, the situation might change for aerosol particles with organic components from the microlayer.

Organic-rich particles might prevail in a highly viscous or glassy state at low temperature, with the result that there is a competition between water uptake and deposition ice nucleation on the glassy, solidified organic surface (Reid et al., 2018). The effect of kinetic limitations of water diffusion and its impact on equilibration timescales and modes of ice nucleation have already been investigated in various computational studies with model organic substances (e.g. Berkemeier et al., 2014; Lienhard et al., 2015; Price et al., 2015; Fowler et al., 2020). For example, Price et al. (2015) have modelled equilibration times for α-pinene secondary organic material based on experimental diffusion measurements. At temperatures of 260 K and above, these timescales were faster than 1 s for the considered RH range between 5 and 95%. At 240 K, the response time was already in the range of a couple of seconds for low RH values, and might further increase up to hours at upper-tropospheric temperatures (Price et al., 2015). In the AIDA chamber, the hygroscopic behaviour of the particles from the Arctic microlayer and surface seawater samples was unchanged compared to purely inorganic sea salt particles, showing that there were no kinetic limitations with respect to water uptake on the experimental time scale. This excludes deposition nucleation as the relevant ice formation mode and could be a factor contributing to the difference of our ice nucleation results with those from the two CFDC studies by Wilson et al. (2015) and Wolf et al. (2020). The different ice nucleation ability observed in these studies might be caused by a different nucleation mechanism. Wolf et al. (2020) have underlined the highly biologically productive environment of the Eastern Tropical North Pacific Ocean sampling location. If organic-rich particles from this location are probed in a CFDC with a short residence time, liquefaction might be incomplete and deposition nucleation on the solid-like organic surface might be the dominant ice nucleation pathway. The important question that could be dealt with in the proposed intercomparison workshop is whether the same behaviour would also have been observed in the AIDA chamber

experiments with longer observation times. Previous ice nucleation measurements with highly-viscous secondary organic aerosol particles from the oxidation of α-pinene confirm that this is a relevant question (Ignatius et al., 2016; Wagner et al., 2017). Whereas SPIN-CFDC measurements detected a pronounced heterogeneous ice nucleation mode for such particles at about 235 K (Ignatius et al., 2016), AIDA expansion runs conducted at the same temperature only showed ice formation at the homogeneous freezing limit, demonstrating the liquefaction of the particles on the experimental time scale (Wagner et al.,

2017). Moreover, Ladino et al. (2014) have shown that precooling is a factor that controls the ice nucleation ability of highly viscous organic aerosol particles. Precooling led to a decrease in the particles' ice nucleation onsets, presumably because the particles were more viscous or solid-like. Another subject of the proposed intercomparison could be exposing the particles to various RH and temperature conditions prior to CFDC sampling and examining the associated effect on their ability to nucleate ice.

Even if we have found that the dried particles from our microlayer samples showed no notable change in their hygroscopic behaviour compared to inorganic sea salt, we are still far from formulating this as a general statement. It has recently been shown that wintertime Arctic SSA particles originating from open leads featured particularly high volume fractions of organic material that was produced as a cryoprotectant (Kirpes et al., 2019). The organic components formed a thick coating layer around the sea salt core. From a mechanistic point of view, it would be highly interesting to investigate the water uptake and

ice nucleation behaviour of such internally-mixed SSA particles. Apart from experiments with cloud chambers and CFDCs, the direct observation of ice nucleation on individual particles with an environmental scanning electron microscope would be another promising approach (Zimmermann et al., 2007; Wang et al., 2016).

**4.5 Influence of inorganic salts and concluding remarks**

If the SSA particles temporarily encounter a relative humidity below about 40% during their transport in the atmosphere and

760 the precipitation of the inorganic salts, in particularly NaCl, is induced, they can become very efficient INPs at temperatures below 220 K, irrespective of the amount of potentially ice-active organic material that is contained in them (Fig. 5). We found little variation of this strong immersion freezing mode associated with the crystalline salt constituents throughout our investigated microlayer and surface seawater samples (Table 4). However, an important point is the temperature where the SSA particles encounter the low relative humidity. On the one hand, efflorescence at low temperatures might be inhibited if

the SSA particles contain a high volume fraction of organic material (Bodsworth et al., 2010). On the other hand, if efflorescence in not inhibited, the temperature controls which crystalline form of NaCl precipitates from the solution droplets. Below $240 \pm 5$ K, the formation of sodium chloride dihydrate (NaCl $\cdot$ 2H$_2$O) is favoured over anhydrous NaCl (Wagner et al., 2012; Wise et al., 2012; Peckhaus et al., 2016). AIDA ice nucleation experiments with partly deliquesced inorganic sea salt particles that contained a solid core of NaCl $\cdot$ 2H$_2$O instead of anhydrous NaCl showed that the temperature limit below which

the immersion freezing mode of the particles initiated was shifted to much higher temperatures (Wagner et al., 2018). In the case of NaCl $\cdot$ 2H$_2$O, a prominent immersion freezing mode that started at $S_{ice} = 1.35$ was already observed in an expansion

run conducted at 229 K, i.e., at a starting temperature where the partially deliquesced particles that were initially dried at room temperature and thus contained anhydrous NaCl as the solid core only showed ice formation at the homogeneous freezing limit (see Fig. 3a). Low temperature crystallization, if it is not inhibited, would thus increase the range of temperatures where purely inorganic sea salt particles are very active INPs.

The influence of the salt constituents is another example that there are manifold factors which control the ice nucleation activity of SSA particles at cirrus conditions. The activity is not only related to the amount and identity of the organic material that is transferred to the particle phase by the bubble bursting process, but also to the potential crystallisation of NaCl or NaCl $\cdot$ 2H$_2$O and the spatial distribution of organic and inorganic components in the particles, which could affect the mode of the ice nucleation process. As such, our measurements add a piece of new information to this research topic, but we are still far from developing a generalised parameterisation for the SSA particles' ice nucleation ability under cirrus conditions, for which also the RH history of the particles would be a crucial factor. In summary, we have shown that partly deliquesced particles from our surface seawater samples and the SM100 diatom culture are very efficient INPs below 220 K due to immersion freezing induced by the solid NaCl core, but have found a much lower heterogeneous ice nucleation activity for the fully deliquesced particles above 220 K, which could be ascribed to immersion freezing by the organic constituents. The latter result deviates from previous ice nucleation measurements with particles generated from microlayer suspensions from other sampling locations. Whether this difference is only a reflection of the strongly varying ice nucleation activity that is commonly also observed under mixed-phase cloud conditions for different microlayer samples, or is also related to the employed ice nucleation measurement techniques, has to be addressed in future intercomparison workshops. The results from such an instrument intercomparison would also be beneficial for the development of parameterisation schemes and for the interpretation of previous and future field measurements at sites which are influenced by marine air masses (Ladino et al., 2016; China et al., 2017).

## Appendix A – The Kongsfjorden sampling site

The high-latitude glacial fjord Kongsfjorden (79°N) is situated at the west coast of the Svalbard Archipelago and has become an established reference site for investigations of the marine ecosystem of the European Arctic (Hop et al., 2002; Svendsen et al., 2002; Wiencke and Hop, 2016). It is affected by both Atlantic and Arctic water masses. Whereas the inner fjord is strongly influenced by glacial run-off from large tidal glaciers, the magnitude of glacial effects is reduced towards the outer fjord, where advection of warm and saline Atlantic water by the West Spitsbergen Current (WSC) is an important factor to control the biological activity and diversity. The spring phytoplankton bloom typically starts by the end of April, but relatively high production levels are maintained during the summer season, which is characterised by diverse phytoplankton communities (Hop et al., 2002; Iversen and Seuthe, 2011; van de Poll et al., 2018).

The seawater samples for this study were collected on July 5, 2017 in the transitional zone of Kongsfjorden, east of the settlement of Ny-Ålesund in possible influence of the outflow from the Midtre Lovénbreen glacier. The sampling took place on a small inflatable boat (Zodiac) using a Niskin bottle sampler placed horizontally onto the water surface. The pre-cleaned Niskin sampler was triple-rinsed with sample water prior to sample collection. The surface seawater was filled from the Niskin sampler outlet directly into sterile sampling bags (Whirl-Pak®, Roth) and stored at -20°C for the chemical and INP analyses. The weather was windy and cloudy and the sea was rough. Wind speeds measured at the nearby meteorological station of Ny-Ålesund ranged between 3.4 and 6.2 m/s, wind directions between 244 and 272°, and the air temperature was about 4°C during the sampling period (Maturilli, 2018).

Table A1 summarises the measured aquatic chemistry and bacterial abundance of the samples. For the measurement of the aquatic chemistry, a portion of each sample was filtered through a precombusted glass fibre filter (MN GF-5, Macherey-Nagel, 25 mm). Anions and cations were determined by ion-chromatography (ICS-1100 respectively ICS-1000, Dionex). For the determination of dissolved organic carbon (DOC) (TOC-CPH, Shimazdu) and dissolved nitrogen (DN) (TNM-1, Shimazdu), the samples were acidified with 200 µL 2N HCl to reach a pH of 1.5 to 2. For the determination of the bacterial abundance, 45 mL of liquid were filled to sterile 50 mL falcon tubes (Roth) and fixated with 2.5 mL formol (formaldehyde, 35% sterile). The fixated samples were then filtrated (Maine Manufacturing, Polycarbonate, black, 0.2 µm, 25 mm diameter), pigmented with 100 µL DAPI (4'6-Diamidino-2-phenylindol, Roth), and affixed to an object plate with immersion oil (Cargille, Type A) after 5–7 minutes of exposure time. The bacterial abundance, $N_{bac}$, was determined with an epifluorescence microscope (Axiophot 2, Zeiss), with a total amplification of 1000 – 1600 using Eqs. A1 and A2.

$$N_{bac} = \frac{N \cdot A_{tot} \cdot DF}{A_{ctd} \cdot V} \tag{A1}$$

$$DF = \frac{V_{tot} + V_{fix}}{V_{tot}} \tag{A2}$$

Here, $N_{bac}$ denotes the number of bacteria (bacteria per mL), $N$ the number of counted bacteria, $A_{tot}$ the total area of the filter ($346 \cdot 106$ µm$^2$), $DF$ the dilution factor, $A_{ctd}$ the counted area of the filter (µm$^2$), $V$ the filtrated volume (mL), $V_{tot}$ the total volume of the sample (mL), and $V_{fix}$ the volume of added fixative (mL).

**Data availability**

Upon manuscript acceptance, we will archive the data derived in this work in the KITopen repository, the central publication platform for KIT (Karlsruhe Institute of Technology) scientists (Open Access, contact: KITopen@bibliothek.kit.edu), and assign them a citable persistent identifier (DOI).

## Author contributions

Conceptualisation: LI, MES, and RW. Investigation: RW, NE, NSU, and OM. Resources: AKB, NE, EG, BJM, and MES. Visualisation: RW and BJM. Writing – original draft preparation: RW and NE. Writing – review & editing: all authors.

## Competing interests

The authors declare that they have no conflict of interest.

## Acknowledgments

We gratefully acknowledge the continuous support by all members of the Engineering and Infrastructure group of IMK-AAF, in particular by Olga Dombrowski, Rainer Buschbacher, Tomasz Chudy, Steffen Vogt, and Georg Scheurig. This work has been funded by the Helmholtz-Gemeinschaft Deutscher Forschungszentren as part of the program "Atmosphere and Climate". Luisa Ickes was supported by the Swiss National Science Foundation (Early Postdoc.Mobility). Benjamin J. Murray was supported by the European Research Council (MarineIce; grant no. 648661). Nsikanabasi Silas Umo was supported by Alexander von Humboldt Foundation, Germany (grant no. 1188375). Matthew E. Salter was supported by the Swedish Research Council (grant no. 2016-05100).

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

**Table 1:** Alphabetical list of previous ice nucleation measurements with SSA particles and their proxies at cirrus conditions, specifying the investigated substances and the employed ice nucleation measurement techniques.

| Study | Substances | Techniques |
|---|---|---|
| Alpert et al. (2011a) | Intact and fragmented cells of *Nannochloris atomus* and *Emiliania huxleyi* | Deposition nucleation: Temperature- and humidity-controlled environmental cell with intact diatoms and fragments of diatoms deposited on a hydrophobic surface<br>Immersion freezing: Aerosol conditioning cell with aqueous NaCl/diatom droplets coupled to cryo-cooling stage |
| Alpert et al. (2011b) | Intact and fragmented cells of *Thalassiosira pseudonana* | See Alpert et al. (2011a) |
| Knopf et al. (2011) | Intact and fragmented cells of *Thalassiosira pseudonana* | See Alpert et al. (2011a) |
| Kong et al. (2018) | Inorganic sea salt | Spectrometer for Ice Nuclei – Continuous Flow Diffusion Chamber (SPIN-CFDC) |
| Ladino et al. (2016) | Exudates from cultures of *Thalassiosira pseudonana*, *Nannochloris atomus*, *Emiliania huxleyi*, and *Vibrio harveyi* | University of Toronto – Continuous Flow Diffusion Chamber (UT-CFDC) |
| Schill and Tolbert (2014) | Inorganic sea salt | Temperature- and humidity-controlled environmental cell equipped with Raman spectrometer |
| Wagner et al. (2018) | Inorganic sea salt | Expansion cooling in the Aerosol Interaction and Dynamics in the Atmosphere (AIDA) cloud chamber |
| Wilson et al. (2015) | Sea surface microlayer samples from the North Pacific and the British Columbia coastline; exudates from a culture of *Thalassiosira pseudonana* | UT-CFDC for the microlayer samples and immersion freezing technique of Alpert et al. (2011a) for the exudates |
| Wolf et al. (2019) | Culture of *Prochlorococcus* | SPIN-CFDC |
| Wolf et al. (2020) | Sea surface microlayer samples from the Eastern Tropical North Pacific Ocean and the Florida Straits | SPIN-CFDC |


**Table 2:** Overview of the investigated samples, specifying the sampling location, sampling time, and coordinates. The last column denotes the median equal-volume sphere diameter, $d_v$, of the particles that were generated by nebulising the bulk solutions and then injected into to AIDA chamber for the ice nucleation experiments.

| Sample name | Location | Sampling time (UTC) | Coordinates | $d_v$ (μm) |
|---|---|---|---|---|
| STN1 | Canadian Arctic | 2016/07/20 16:30 | 60°17.921'N, 62°10.750'W | 0.70 |
| STN2 | Canadian Arctic | 2016/07/29 15:30 | 67°23.466'N, 63°22.067'W | 0.75 |
| STN7 | Canadian Arctic | 2016/08/11 17:00 | 77°47.213'N, 76°29.841'W | 0.65 |
| SML6 | Greenland Sea | 2013/07/22 08:00 | 73°06.340'N, 13°06.120'W | 0.75 |
| SML10 | Greenland Sea | 2013/07/26 14:24 | 76°16.141'N, 05°18.642'W | 0.75 |
| SML12.5 | Greenland Sea | 2013/07/29 08:50 | 77°27.207'N, 05°13.610'W | 0.75 |
| SML13 | Greenland Sea | 2013/07/30 12:01 | 74°48.828'N, 07°35.043'E | 0.80 |
| KFJ1 | Kongsfjorden | 2017/07/05 12:44 | 78°55.556'N, 12°02.496'E | 0.75 |
| KFJ2 | Kongsfjorden | 2017/07/05 12:58 | 78°55.508'N, 12°03.282'E | 0.75 |
| KFJ3 | Kongsfjorden | 2017/07/05 12:10 | 78°55.447'N, 12°03.994'E | 0.80 |
| KFJ4 | Kongsfjorden | 2017/07/05 13:18 | 78°55.381'N, 12°04.435'E | 0.75 |
| KFJ5 | Kongsfjorden | 2017/07/05 13:27 | 78°55.366'N, 12°04.883'E | 0.75 |
| SM100 | Laboratory sample | - | - | 0.70 |


**Table 3:** Quantitative analysis of the AIDA expansion cooling runs started at 229 K. As discussed in the text, the observed heterogeneous ice nucleation modes were separated into two nucleation ranges: (i) $S_{ice} = 1.10 - 1.38$ and (ii) $S_{ice} = 1.38 - 1.48$. For both ranges, the maximum values of the ice-active fraction of the aerosol particles, $FF$, and the ice nucleation active surface site density, $n_s$, were evaluated. The last column summarises the $T_{50}$ freezing temperatures from the cold-stage freezing measurements with 50 μL aliquots of the bulk solutions.

| Sample | Nucleation range $S_{ice} = 1.10 - 1.38$ | | Nucleation range $S_{ice} = 1.38 - 1.48$ | | Bulk freezing |
|---|---|---|---|---|---|
| | $FF$ (%) | $n_s$ ($10^7$ m$^{-2}$) | $FF$ (%) | $n_s$ ($10^8$ m$^{-2}$) | $T_{50}$ (K) |
| STN1 | 0.006 | 4.0 | 0.05 | 3.3 | 256.5 |
| STN2 | 0.01 | 6.0 | 0.12 | 7.2 | 265.7 |
| STN7 | < 0.001 | < 0.5 | 0.003 | 2.0 | 260.3 |
| SML6 | 0.01 | 5.6 | 0.08 | 4.5 | 263.0 |
| SML10 | 0.008 | 4.5 | 0.07 | 4.0 | 261.9 |
| SML12.5 | 0.003 | 1.7 | 0.07 | 4.0 | 263.3 |
| SML13 | 0.001 | 0.5 | 0.22 | 11.1 | 258.3 |
| KFJ1 | < 0.001 | < 0.5 | 0.003 | 0.2 | 263.2 |
| KFJ2 | 0.003 | 1.7 | 0.02 | 1.1 | 262.7 |
| KFJ3 | 0.001 | 0.5 | 0.004 | 0.2 | 261.9 |
| KFJ4 | 0.01 | 5.4 | 0.02 | 1.1 | 262.3 |
| KFJ5 | 0.01 | 5.6 | 0.06 | 3.4 | 262.0 |
| SM100 | 0.003 | 1.8 | 0.66 | 39.6 | 254.7 |
| SM100_dil | < 0.001 | < 0.5 | 0.04 | 2.4 | – |

**Table 4:** Quantitative analysis of the AIDA expansion cooling runs started at 217 K. The sample called "blank" refers to a commercial bulk Atlantic water sample that has previously been probed in the AIDA chamber (Wagner et al., 2018). Due to the similarity of the ice nucleation behaviour between the samples, not all of them were probed at this temperature (no experiment with STN1, KFJ4, KFJ5, and SM100_dil). $T_0$: Start temperature of the expansion cooling run. $T_{ice,onset}$: Temperature at the onset of ice nucleation ($FF = 0.1\%$). $S_{ice,onset}$: Ice saturation ratio at the onset of ice nucleation. $T_{ice,max}$: Temperature when the maximum ice saturation ratio during the expansion run was reached. $S_{ice,max}$: Maximum ice saturation ratio during the expansion run. $n_s(S_{ice,max})$: Ice nucleation active surface site density evaluated at the maximum ice saturation ratio.

| Sample | $T_0$ (K) | $T_{ice,onset}$ (K) | $S_{ice,onset}$ | $T_{ice,max}$ (K) | $S_{ice,max}$ | $n_s(S_{ice,max})$ ($10^{10}$ m$^{-2}$) |
|--------|-----------|---------------------|-----------------|-------------------|----------------|------------------------------------------|
| Blank | 217.7 | 215.4 | 1.24 | 214.7 | 1.30 | 5.2 |
| STN2 | 216.7 | 214.4 | 1.26 | 213.7 | 1.35 | 5.3 |
| STN7 | 216.7 | 214.5 | 1.27 | 213.8 | 1.33 | 6.2 |
| SML6 | 216.6 | 214.6 | 1.26 | 214.1 | 1.30 | 4.9 |
| SML10 | 216.7 | 214.6 | 1.27 | 213.8 | 1.32 | 6.0 |
| SML12.5 | 216.7 | 214.4 | 1.28 | 213.7 | 1.33 | 5.3 |
| SML13 | 216.7 | 214.5 | 1.27 | 213.9 | 1.32 | 5.4 |
| KFJ1 | 216.7 | 214.7 | 1.25 | 213.9 | 1.32 | 5.5 |
| KFJ2 | 216.6 | 214.6 | 1.26 | 213.8 | 1.32 | 5.4 |
| KFJ3 | 216.6 | 214.5 | 1.27 | 214.2 | 1.31 | 5.0 |
| SM100 | 217.1 | 214.5 | 1.28 | 213.7 | 1.33 | 5.7 |

**Table A1:** Aquatic chemistry and bacterial abundance, $N_{bac}$, of the five surface seawater samples from Kongsfjorden. See Appendix A for details on the analytical methods.

| Sample | DOC (μg/L) | DN (μg/L) | $Cl^-$ (mg/L) | $NO_3^-$ (mg/L) | $SO_4^{2-}$ (mg/L) | $Na^+$ (mg/L) | $NH_4^+$ (mg/L) | $K^+$ (mg/L) | $Mg^{2+}$ (mg/L) | $Ca^{2+}$ (mg/L) | $N_{bac}$ (#/mL) |
|---|---|---|---|---|---|---|---|---|---|---|---|
| KFJ1 | 1523 | 558.9 | 17508.7 | 8.9 | 2890.1 | 10137.9 | 33.0 | 334.6 | 1091.4 | 312.9 | $3.1 \cdot 10^4$ |
| KFJ2 | 1092 | 193.3 | 18438.2 | 10.2 | 2636.8 | 10421.2 | 4.2 | 340.0 | 1192.4 | 371.0 | $2.2 \cdot 10^4$ |
| KFJ3 | 1294 | 251.6 | 19066.6 | 10.5 | 2525.7 | 10746.8 | 14.1 | 350.2 | 1233.2 | 389.7 | $7.4 \cdot 10^4$ |
| KFJ4 | 1152 | 131.4 | 19215.4 | 10.2 | 2536.7 | 10847.5 | 21.0 | 351.3 | 1232.5 | 394.5 | $8.1 \cdot 10^4$ |
| KFJ5 | 1086 | 94.7 | 19110.7 | 10.1 | 2520.2 | 10760.2 | 2.5 | 352.4 | 1241.6 | 387.6 | $3.5 \cdot 10^4$ |


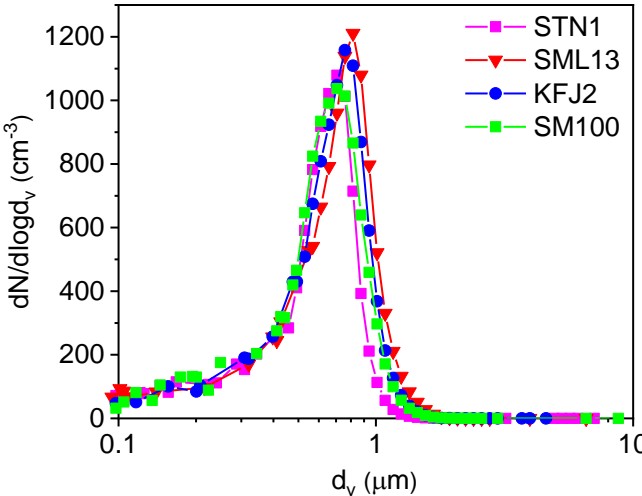

**Figure 1:** Number size distributions of aerosol particles generated by nebulising the STN1, SML13, KFJ2, and SM100 bulk solutions. The size spectra were measured at 298 K and RH < 3%, thereby reflecting the dry particle diameters. The mobility and aerodynamic diameters from the SMPS and APS measurements were converted into the equal-volume sphere diameter, $d_v$, by assuming a dynamic shape factor of 1.08 (Hinds, 1999) and a particle density of 2.017 g cm$^{-3}$ for sea salt (Zieger et al., 2017).

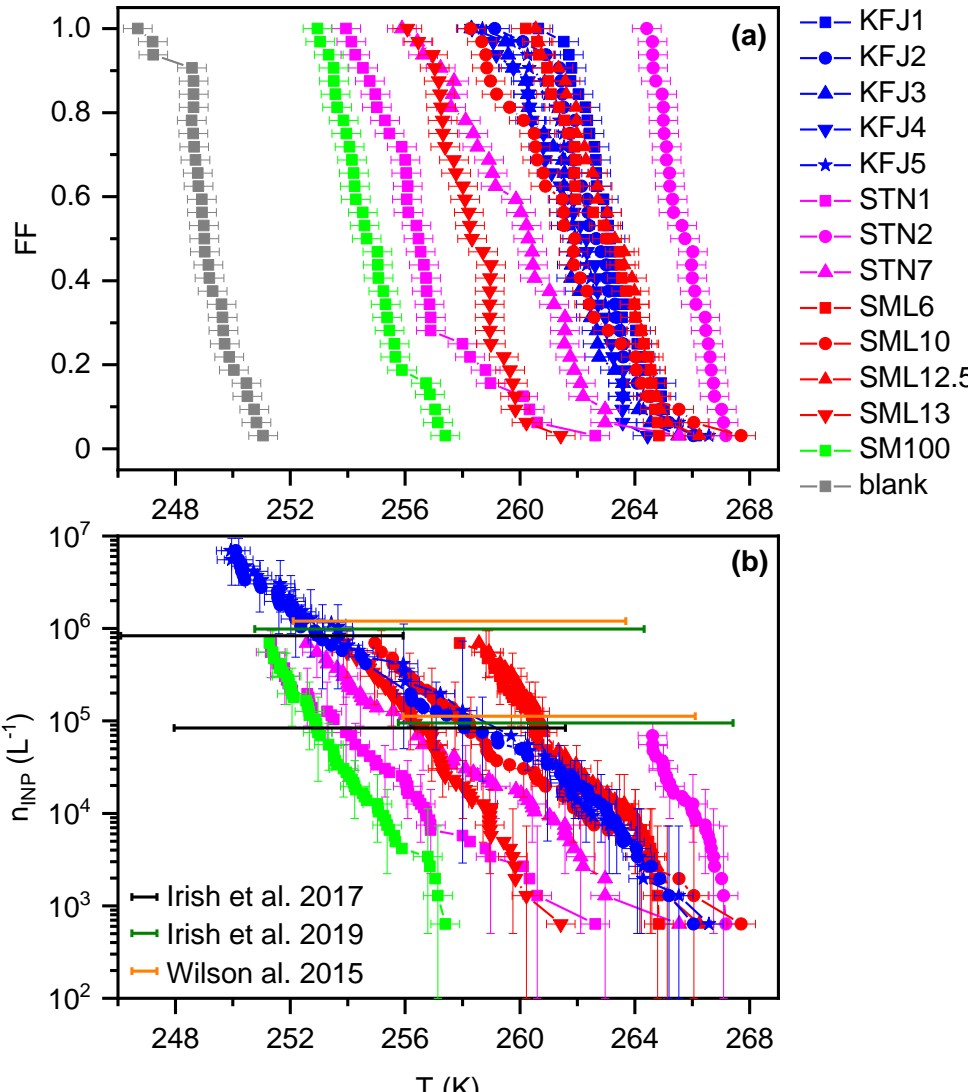

**Figure 2:** (a) Frozen fraction, *FF*, curves of the investigated samples from cold-stage measurements with INSEKT, corrected for the freezing point depression by the salts. (b) Cumulative INP concentrations, $n_{INP}$ (L$^{-1}$), as computed with Eq. (1) from the measurements in panel (a) and additional INSEKT measurements where the solutions were diluted by a factor of 10 and 100 with ultrapure water to extend their $n_{INP}$ (L$^{-1}$) spectrum to lower temperatures. The error bars reflect the statistical uncertainty (Koop et al., 1997). The coloured horizontal bars, positioned at $n_{INP} = 10^5$ and $10^6$ L$^{-1}$ (shown with a slight vertical offset as a matter of clarity), comprise the range of temperatures where such INP concentrations were encountered in previous droplet freezing experiments with Arctic microlayer samples (Wilson et al., 2015; Irish et al., 2017; Irish et al., 2019).

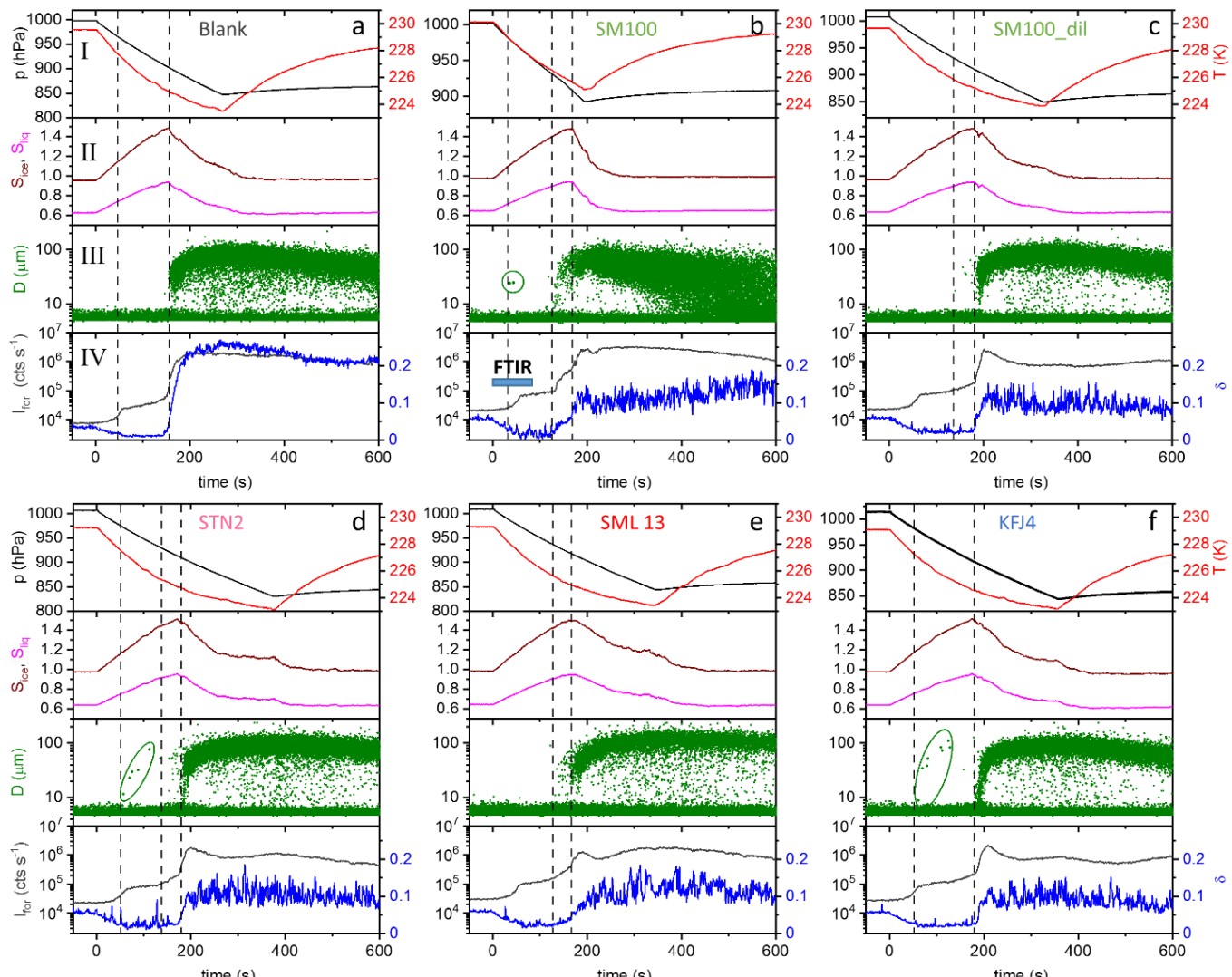

**Figure 3:** Time series of the AIDA records from the expansion cooling experiments started at 229 K for six different samples (a – f). The sample called "blank" refers to a commercial bulk Atlantic water sample that has already previously been probed in the AIDA chamber (Wagner et al., 2018). The four individual panels for each part show the following data: I: AIDA mean gas temperature (red line) and pressure (black line); II: saturation ratios with respect to ice ($S_{ice}$, brown line) and supercooled water ($S_{liq}$, magenta line); III: size-resolved single-particle scattering signals from the optical particle counter (green dots); IV: forward scattering intensity ($I_{for}$, 2° scattering angle, grey line) and backscattering linear depolarization ratio ($\delta$, 178° scattering angle, blue line). The blue horizontal bar in panel IV of Fig. 3b symbolises the time period where the series of FTIR spectra shown in Fig. 4 were recorded. See text for details.

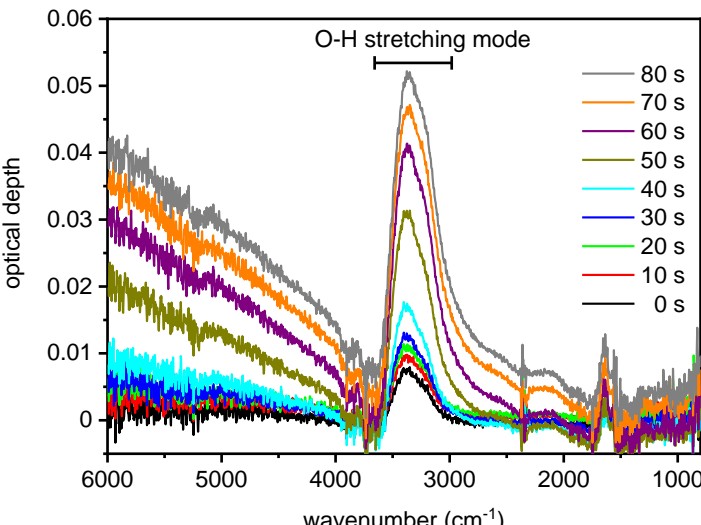

**Figure 4:** Series of infrared extinction spectra that were recorded in the first 80 s during the expansion run with the particles generated from the SM100 culture (see Fig. 3b). The broad peak between about 3500 and 2800 cm$^{-1}$ is due to the O – H stretching mode of liquid water. As described in the text, this spectra series is representative for all measurements with the various samples and is therefore also used to discuss the hygroscopic behaviour of the particles from the bulk Atlantic water sample in Sect. 3.2.1.

1340

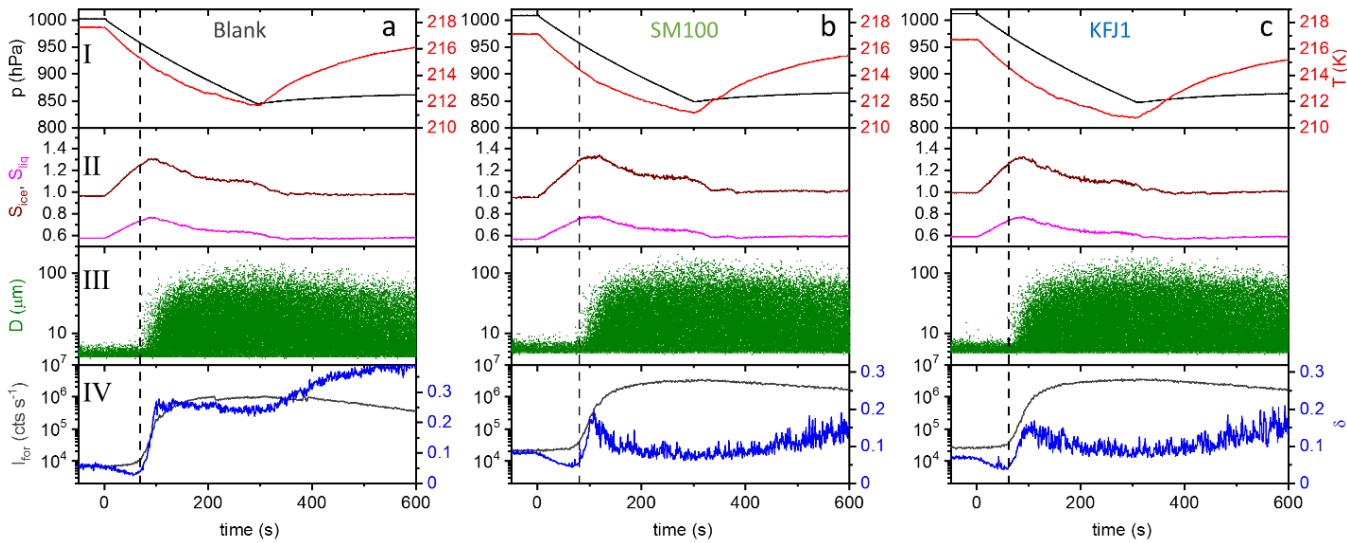

1345

**Figure 5:** Time series of the AIDA records from the expansion cooling experiments started at 217 K for three different samples (a – c). The sample called "blank" again refers to the commercial bulk Atlantic water sample that has already previously been probed in the AIDA chamber (Wagner et al., 2018). The individual panels show the same data types as in Fig. 3.

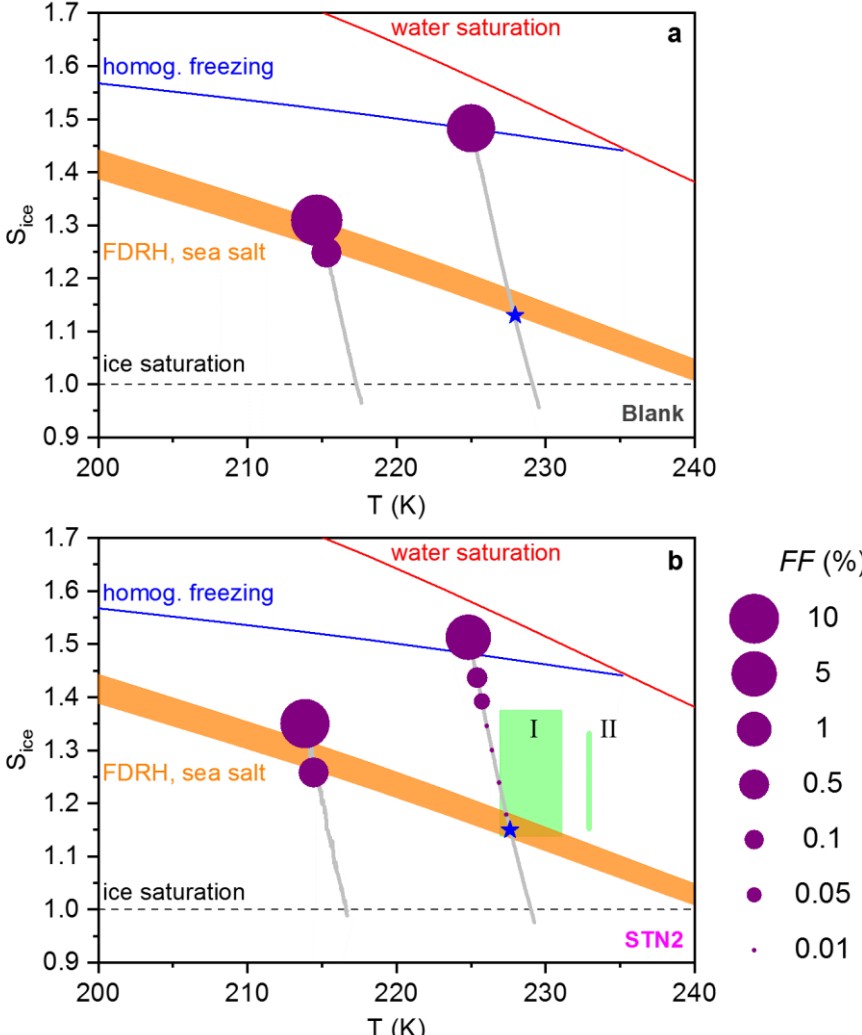

**Figure 6:** Illustration of the ice nucleation behaviour observed during the AIDA expansion runs with the particles generated from the bulk Atlantic water sample (blank, a) and the STN2 microlayer sample (b). The $S_{ice}$ vs. $T$ trajectories of the AIDA experiments are shown as the grey lines. The dashed black line denotes ice-saturated conditions, the red line displays the saturation curve with respect to supercooled liquid water, the homogeneous freezing onset of aqueous solution droplets is indicated by the blue line (Koop et al., 2000b), and the shaded orange area shows an estimate for the RH range of the full deliquescence (FDRH) of inorganic sea salt particles as discussed in the text. Blue stars denote measured FDRH onsets using the SIMONE light scattering data. The size of the purple-coloured dots superimposed on the trajectories represents the $FF$ values encountered during the expansion runs. The green box labelled I denotes the range of onset conditions ($FF = 1\%$) for ice nucleation on the particles from the microlayer samples investigated by Wolf et al. (2020) (Eastern Tropical North Pacific Ocean and Florida Straits, temperature range 227 – 231 K). The green bar labelled II represents the range of ice nucleation onsets ($FF = 1\%$) observed in the experiments by Wilson et al. (2015) (aerosolised microlayer samples from the North Pacific and the British Columbia coastline probed at 233 K).

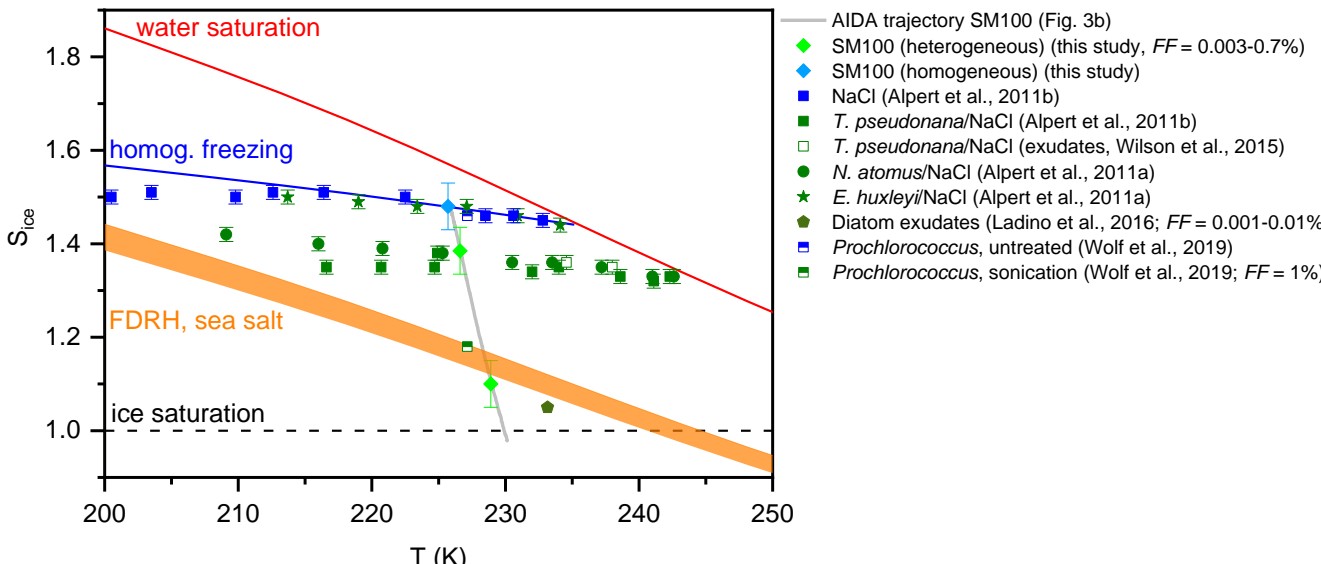

**Figure 7:** Homogenous and heterogeneous ice nucleation onsets from the AIDA expansion run with the particles generated from the SM100 culture shown in Fig. 3b (light green and blue diamonds) in comparison with data from previous experiments with phytoplankton and marine bacterial species (see Sect. 1.2). The onsets from the immersion freezing measurements with the technique by Alpert et al. (2011a) were evaluated at median freezing temperatures for homogeneous and heterogeneous ice nucleation. The heterogeneous ice nucleation onsets from the AIDA and the various CFDC experiments correspond to different *FF* values as indicated in the legend and discussed in Sect. 4.1. The $S_{ice}$ vs. *T* trajectory of the AIDA experiment is shown as the grey line. The other reference lines are the same as shown in Fig. 6.