# Peer review of "Heterogeneous ice nucleation ability of aerosol particles generated from Arctic sea surface microlayer and surface seawater samples at cirrus temperatures"

_Atmospheric Chemistry and Physics, 2021_

## Referee Comment (RC1)

**Heterogeneous ice nucleation ability of aerosol particles generated from Arctic sea surface microlayer and surface seawater samples at cirrus temperatures**

Robert Wagner, Luisa Ickes, Allan K. Bertram, Nora Els, Elena Gorokhova, Ottmar Möhler, Benjamin J. Murray, Nsikanabasi Silas Umo, and Matthew E. Salter

https://doi.org/10.5194/acp-2021-252

**Summary**
The authors present new data on the ice nucleation ability of aerosol generated from Arctic seawater samples. Further experiments investigated the ice nucleation ability of aerosols comprised of inorganic sea salt and aerosols generated from a culture of the diatom *Skeletonema marinoi*. Using active site densities, conditions at the onset of nucleation, and median freezing temperatures, the team quantified ice nucleation in both the mixed-phase cloud regime (~248-268 K) using off-line droplet freezing techniques and in the cirrus cloud regime (~210–235 K) using the AIDA chamber. The ice nucleation abilities of particles was compared between the mixed-phase cloud and cirrus cloud regime.

**General Comments**
The data presented here make a notable contribution to the field of aerosol-cloud interactions by systematically investigating the ice nucleation abilities of sea-spray analogues at cirrus-relevant conditions. Although INPs derived from Arctic seawater have been the topic of several recent publications, few have studied the ice nucleation abilities at temperatures cold enough to speak to their potential impact on cirrus cloud properties. Although ice nucleation in the cirrus regime is less frequently studied than in the mixed-phase cloud regime, the climatic impact of cirrus clouds means aerosol-cloud interactions at cirrus temperatures warrant further exploration in studies such as this one.

The authors should be commended for investigating a relationship between the ice nucleation abilities of seawater-derived particles in the mixed-phase and cirrus cloud regimes. This is an interesting addition that I hope future studies will expand on. The authors should also be commended for their efforts to compare their results to previous studies and discuss possible scientific and methodological reasons for differences.

I therefore support the publication of this manuscript in ACP pending minor revisions. Below, I outline some requested revisions, while also providing a few questions and comments to clarify some of the text's conclusions.

**Primary Comments**
1. A note on ice nucleation terminology — I encourage the authors to consider whether "immersion freezing" is truly the best term to describe the process of ice nucleation upon deliquescence (or in pre-deliquescence hygroscopic growth) but below liquid water saturation in the cirrus cloud regime.
    a. Although I acknowledge that other publications have referred to this process as "immersion freezing" — and I do not insist that an alternative term be adopted here — it is somewhat unintuitive to refer to freezing processes below water saturation as immersion freezing. The authors themselves allude to my concerns in line 101: "Note, immersion freezing in this case is different to immersion freezing under mixed-phase cloud conditions since it occurs below liquid water saturation."
    b. As a possible alternative, I might suggest the term deliquescence-freezing, as proposed by Khvorostyanov et al., 2004.

2. The authors present data seeming to indicate that at cold temperatures (<217 K), the fractional ice activation of inorganic sea salt increases after reaching the full deliquescence relative humidity. (I refer to the data in Figure 6a.)
   a. This is counterintuitive to me, as I would expect full deliquescence of inorganic salt particles to result in a totally aqueous solution that would preclude further particles from heterogeneously nucleating ice. Can the authors please clarify or explain this in the text?

3. The differences between the present study's findings and those from Wilson et al. 2015 and Wolf et al. 2020 are interesting. I would like to see a little more exploration of the oceanographic reasons as to why the results from this study (from the Arctic) do not follow the same patterns as those from Wilson et al. and Wolf et al (from the tropics).
   a. For example: were the average rates of primary productivity, and therefore perhaps the organic content of the seawater, higher in the tropics? And if so, could this help to explain the observed higher $n_S$ values for the seawater from Wilson et al.'s and Wolf et al.'s analyses? This could be easily explored by comparing satellite retrievals of surface chlorophyl-a concentrations.

4. The authors propose that the ice nucleation measurement technique might contribute to the differences between their results and the results of Wilson et al. and Wolf et al. Specifically, the authors suggest that the differences in temperature and relative humidity trajectories experienced by aerosol particles in the AIDA and CFDC techniques may impact ice nucleation behaviour. The discussion covering this (Section 4.4) is rather speculative. It could be strengthened with further citations and details on key points the authors touch on. I outline a few specific recommendations below:
   a. The authors should better reference the literature on temperature and relative humidity trajectories as particles enter CFDCs. For example, I recommend citing and discussing the results in Rogers, 1988, Garimella et al., 2016, and Kulkarni and Kok, 2012. These papers include simulations describing the T and RH trajectories experienced by particles as they enter the CFDC.
   b. Once the T and RH trajectories are better discussed, the authors should expand on how the differences between CFDCs and AIDA can lead to physical differences in particle water uptake. The authors should mention that the rates of water uptake through organic coatings can be estimated for model organics (e.g. Price et al., 2015; Renbaum-Wolff et al., 2013). These calculations, even if not yet possible for complex mixtures of marine organics, at least provide a theoretical underpinning for the types of issues the authors raise.
   c. The authors might expand on their call for an intercomparison by suggesting that part of the intercomparison could be to subject aerosols to various "precooling" trajectories, i.e., controlling the temperature and relative humidity of aerosols in a large mixing chamber (NAUA?), prior to CFDC sampling. This could help facilitate comparison between AIDA and CFDC data.
   d. The manuscript is already quite long, and I do not mean to require that the authors add much text in response to the above points. Two or three sentences for each point should suffice.

**Minor Comments**
**Abstract**
5. Line 16: "*Only a small fraction of sea salt aerosol is transported to the upper troposphere...*" Please change "sea salt aerosol" to "sea spray aerosol," since these aerosol particles are often internally mixed and compositionally complex, consisting of more than just salts.

6. Line 22: "*The particles were suspended in a large cloud chamber...*" I think it would be useful to specifically mention AIDA here.

7. Line 32: "*we also discuss how far instrumental parameters...*" Semantical point, but I feel you don't discuss "har far" – i.e., quantify the extent to which – these parameters might impact results. You only discuss that they could impact the results without supplying an estimate for how large the magnitude of the impact might be. I would remove the word "far."

**Introduction**

8. Line 44: "*…homogeneous freezing of pure water droplets, which takes place below about 235 K.*" Please cite Koop et al. 2000 here, or your reference of choice.

9. Line 63: "*…the freezing data are usually reported as the temperature-dependent number of INPs per either droplet volume or volume of collected air.*" Temperature-dependence is reported for INP concentrations in the mixed-phase cloud regime; but in the cirrus regime below liquid water saturation, INP concentration is reported as a function of both temperature \*and\* relative humidity. Please clarify this in the text.

10. Line 78: "*…showed contributions of up to 25% from sea salt over ocean regions.*" Please change to "over ocean and coastal regions."

11. Line 81: "*…if we are to explain regional indications of heterogeneous ice nucleation activity...*" I'm not sure I understand the meaning of the word "indications" here. Perhaps change to "importance" or "impact" or "variability?"

12. Lines 155-158: Can the authors clarify what is meant by "partial deliquescence" or "before full deliquescence?" Do they refer to pre-deliquescence uptake of water (i.e., hygroscopic growth below the DRH)? Or do they refer to the time between the start of deliquescence and full deliquescence above the DRH?

13. Line 166 and throughout: I think "*Emiliana huxleyi*" should be spelled "*Emilian**ia** huxleyi.*"

14. Line 174 and throughout: I think "*Perchlorococcus*" should be spelled "***Pro**chlorococcus.*"

**Experimental**

15. Line 283: "*…particles generated by nebulising the undiluted microlayer and surface seawater samples…*" were the samples homogenized (e.g., shaken) after thawing and prior to aerosolization?

16. Line 318*: "For a subset of samples, we diluted the suspensions by a factor of 10 and 100 with ultrapure water to extend the measured $n_{INP}(T)$ spectrum to lower freezing temperatures*." At what temperatures was this extra dilution step necessary?

17. Line 342: "*…we calculated the ice nucleation active surface site density, ns, with an estimated uncertainty of ± 40%.*" This uncertainty range seems rather large. Can the authors briefly summarize here the factors that go into calculating uncertainty in nS?

18. Line 345: *"…we consider the extreme scenario…"* Is this considered "extreme" because experimental experience demonstrates that counting frequency is typically much higher?

19. Lines 346-349: This is a nice description of lower limits!

**Results**

20. Line 362: *"…corrected for the freezing point depression by the salts."* Can the authors provide more description, or a reference, as to how this was freezing point depression correction was done?

21. Line 373: "*This may be partly explained by the weather conditions...*" E.g., high winds? Can the author provide a typical wind speed during these measurements? The authors might reference one of the numerous studies indicating the wind speed at which the microlayer breaks up, e.g. Wurl et al., 2011.

22. Line 409: "*…any observable heterogeneous ice nucleation mode must be related to the organic material contained in the aerosol particles because the inorganic salt components are not yet ice-active at this temperature.*" Is it not also possible that the seawater also contains dust? See e.g. Cornwell et al., 2020.

**Discussion and Outlook**

23. Line 574: "…the processing of exudates either through biological processes such as microbial metabolism or physicochemical processes…" Please add a reference to support this discussion of microbial metabolism. I suggest either McCluskey et al., 2017 and/or Wang et al., 2015, but feel free to add another.

24. Line 583: "However, the amount of dispersed ice-nucleating entities was obviously much smaller than in the Wolf et al. (2019) study." Please state whether the cell concentrations were similar or different between this study and Wolf et al.

25. Line 628: "…where the bursting of bubble cap films can lead to the formation of highly organically enriched particles." Please add a reference.

26. Line 631: "*…the ice nucleation mode might change from immersion freezing, as observed in the AIDA experiments, to deposition nucleation, where ice formation initiates by the deposition of water vapour on crystalline or glassy surfaces.*" Please add a reference (or two) that discusses depositional freezing on glassy organic aerosols. E.g. Murray et al., 2010.

27. Line 673: "Its transit time through the nucleation region of a CFDC is typically about 10 seconds." Please add a reference for this residence time. E.g. Garimella et al., 2016.

**Figures and Tables**

Table 1: Be sure to correct the spelling of the names here, as indicated above.

Table 2: What does the uncertainty or variability in the mean diameter (±0.05 μm) represent?

**Works Cited**

Cornwell, G. C., Sultana, C. M., Prank, M., Cochran, R. E., Hill, T. C. J., Schill, G. P., DeMott, P. J., Mahowald, N. and Prather, K. A.: Ejection of Dust From the Ocean as a Potential Source of Marine Ice Nucleating Particles, J. Geophys. Res. Atmos., 125(24), doi:10.1029/2020JD033073, 2020.

Garimella, S., Kristensen, T. B., Ignatius, K., Welti, A., Voigtländer, J., Kulkarni, G. R., Sagan, F., Kok, G. L., Dorsey, J., Nichman, L., Rothenberg, D. A., Rösch, M., Kirchgäßner, A. C. R., Ladkin, R., Wex, H., Wilson, T. W., Ladino, L. A., Abbatt, J. P. D., Stetzer, O., Lohmann, U., Stratmann, F. and Cziczo, D. J.: The SPectrometer for Ice Nuclei (SPIN): an instrument to investigate ice nucleation, Atmos. Meas. Tech., 9(7), 2781–2795, doi:10.5194/amt-9-2781-2016, 2016.

Khvorostyanov, V. I., Curry, J. a., Khvorostyanov, V. I. and Curry, J. a.: The Theory of Ice Nucleation by Heterogeneous Freezing of Deliquescent Mixed CCN. Part I: Critical Radius, Energy, and

Nucleation Rate, J. Atmos. Sci., 61(22), 2676–2691, doi:10.1175/JAS3266.1, 2004.

Kulkarni, G. and Kok, G.: Mobile Ice Nucleus Spectrometer, Pacific Northwest Natl. Lab. Richland, WA, 2012.

McCluskey, C. S., Hill, T. C. J., Malfatti, F., Sultana, C. M., Lee, C., Santander, M. V., Beall, C. M., Moore, K. A., Cornwell, G. C., Collins, D. B., Prather, K. A., Jayarathne, T., Stone, E. A., Azam, F., Kreidenweis, S. M. and DeMott, P. J.: A Dynamic Link between Ice Nucleating Particles Released in Nascent Sea Spray Aerosol and Oceanic Biological Activity during Two Mesocosm Experiments, J. Atmos. Sci., 74(1), 151–166, doi:10.1175/JAS-D-16-0087.1, 2017.

Murray, B. J., Wilson, T. W., Dobbie, S., Cui, Z., Al-Jumur, S. M. R. K., Möhler, O., Schnaiter, M., Wagner, R., Benz, S., Niemand, M., Saathoff, H., Ebert, V., Wagner, S. and Kärcher, B.: Heterogeneous nucleation of ice particles on glassy aerosols under cirrus conditions, Nat. Geosci., 3(4), 233–237, doi:10.1038/ngeo817, 2010.

Price, H. C., Mattsson, J., Zhang, Y., Bertram, A. K., Davies, J. F., Grayson, J. W., Martin, S. T., O'Sullivan, D., Reid, J. P., Rickards, A. M. J. and Murray, B. J.: Water diffusion in atmospherically relevant α-pinene secondary organic material, Chem. Sci., 6(8), 4876–4883, doi:10.1039/C5SC00685F, 2015.

Renbaum-Wolff, L., Grayson, J. W., Bateman, A. P., Kuwata, M., Sellier, M., Murray, B. J., Shilling, J. E., Martin, S. T. and Bertram, A. K.: Viscosity of α-pinene secondary organic material and implications for particle growth and reactivity., Proc. Natl. Acad. Sci. U. S. A., 110(20), 8014–9, doi:10.1073/pnas.1219548110, 2013.

Rogers, D. C.: Development of a continuous flow thermal gradient diffusion chamber for ice nucleation studies, Atmos. Res., 22(2), 149–181, doi:10.1016/0169-8095(88)90005-1, 1988.

Wang, X., Sultana, C. M., Trueblood, J., Hill, T. C. J., Malfatti, F., Lee, C., Laskina, O., Moore, K. A., Beall, C. M., McCluskey, C. S., Cornwell, G. C., Zhou, Y., Cox, J. L., Pendergraft, M. A., Santander, M. V., Bertram, T. H., Cappa, C. D., Azam, F., DeMott, P. J., Grassian, V. H. and Prather, K. A.: Microbial Control of Sea Spray Aerosol Composition: A Tale of Two Blooms, ACS Cent. Sci., 1(3), 124–131, doi:10.1021/acscentsci.5b00148, 2015.

Wurl, O., Wurl, E., Miller, L., Johnson, K. and Vagle, S.: Formation and global distribution of sea-surface microlayers, Biogeosciences, 8(1), 121–135, doi:10.5194/bg-8-121-2011, 2011.

---

## Author Comment (AC1)

**Response to Referee #1**

We thank Referee #1 for the positive evaluation of our article and the detailed comments and suggestions to strengthen and clarify our conclusions. Below, we provide a point-by-point answer to the individual comments (referee report in blue, our answers in black). The page and line numbers refer to the original manuscript.

**Summary**

The authors present new data on the ice nucleation ability of aerosol generated from Arctic seawater samples. Further experiments investigated the ice nucleation ability of aerosols comprised of inorganic sea salt and aerosols generated from a culture of the diatom *Skeletonema marinoi*. Using active site densities, conditions at the onset of nucleation, and median freezing temperatures, the team quantified ice nucleation in both the mixed-phase cloud regime (~248-268 K) using off-line droplet freezing techniques and in the cirrus cloud regime (~210–235 K) using the AIDA chamber. The ice nucleation abilities of particles was compared between the mixed-phase cloud and cirrus cloud regime.

**General Comments**

The data presented here make a notable contribution to the field of aerosol-cloud interactions by systematically investigating the ice nucleation abilities of sea-spray analogues at cirrus-relevant conditions. Although INPs derived from Arctic seawater have been the topic of several recent publications, few have studied the ice nucleation abilities at temperatures cold enough to speak to their potential impact on cirrus cloud properties. Although ice nucleation in the cirrus regime is less frequently studied than in the mixed-phase cloud regime, the climatic impact of cirrus clouds means aerosol-cloud interactions at cirrus temperatures warrant further exploration in studies such as this one.

The authors should be commended for investigating a relationship between the ice nucleation abilities of seawater-derived particles in the mixed-phase and cirrus cloud regimes. This is an interesting addition that I hope future studies will expand on. The authors should also be commended for their efforts to compare their results to previous studies and discuss possible scientific and methodological reasons for differences.

I therefore support the publication of this manuscript in ACP pending minor revisions. Below, I outline some requested revisions, while also providing a few questions and comments to clarify some of the text's conclusions.

**Primary Comments**

1. A note on ice nucleation terminology — I encourage the authors to consider whether "immersion freezing" is truly the best term to describe the process of ice nucleation upon

deliquescence (or in predeliquescence hygroscopic growth) but below liquid water saturation in the cirrus cloud regime.

a. Although I acknowledge that other publications have referred to this process as "immersion freezing" — and I do not insist that an alternative term be adopted here — it is somewhat unintuitive to refer to freezing processes below water saturation as immersion freezing. The authors themselves allude to my concerns in line 101: "Note, immersion freezing in this case is different to immersion freezing under mixed-phase cloud conditions since it occurs below liquid water saturation."

b. As a possible alternative, I might suggest the term deliquescence-freezing, as proposed by Khvorostyanov et al., 2004.

Thank you for bringing up this interesting point of terminology. We think that "immersion freezing" is the most generic description for the observed ice nucleation mode of the mixed solid-liquid sea salt aerosol particles. In the article on ice nucleation terminology by Vali et al. (2015), immersion freezing is defined as ice nucleation initiating within the body of a liquid, which can thus be pure water or an aqueous solution. Let us consider the scenario when atmospheric aging leads to the formation of a coating layer of a liquid (e.g. sulfuric acid) on insoluble INPs like dust or soot. Under cirrus conditions, the included dust and soot cores can induce the heterogeneous freezing of the aqueous sulfuric acid coating before reaching the homogeneous freezing threshold. Immersion freezing would be in such a case the only appropriate term to describe the nucleation mode, because there is no deliquescence step involved. The term "deliquescence freezing", however, is certainly a good description of the specific process that takes place when the initially dry sea salt aerosol particles are probed in an expansion cooling experiment in the AIDA chamber at sufficiently low temperatures, where the particles only partially deliquesce and the yet undissolved core can induce the freezing. We will mention this term on page 5, line 157:

"*The second process has also been termed as "deliquescent-heterogeneous freezing" in the literature (Khvorostyanov and Curry, 2004).*"

2. The authors present data seeming to indicate that at cold temperatures (<217 K), the fractional ice activation of inorganic sea salt increases after reaching the full deliquescence relative humidity. (I refer to the data in Figure 6a.)

a. This is counterintuitive to me, as I would expect full deliquescence of inorganic salt particles to result in a totally aqueous solution that would preclude further particles from heterogeneously nucleating ice. Can the authors please clarify or explain this in the text?

This is a good observation. In our preceding paper on the ice nucleation behaviour of purely inorganic sea salt aerosol particles (Wagner et al., 2018), we have described how we estimated

the full deliquescence relative humidity that is shown in Fig. 6a as the dashed orange line. We thereby referred to the hydration curves of levitated SSA particles measured at 298 K by Tang et al. (1997). Whereas pure NaCl particles just revealed a singular deliquescence transition at 75.3% RH, the multicomponent SSA particles showed a different hygroscopic behaviour. There was a gradual particle growth up to about 71% RH, followed by a very rapid water uptake between 71 and 74% RH (corresponding to the dissolution of the NaCl fraction), to finally yield a fully aqueous solution droplet at 74% RH. To obtain a rough estimate for the *onset* of the full deliquescence step in the SSA particles, i.e., the point where all of the remaining NaCl *starts* to dissolve, we have scaled the extrapolated, temperature-dependent parameterisation of the deliquescence relative humidities of crystalline NaCl particles from Tang and Munkelwitz (1993) with an absolute, temperature-independent shift of –4% on the relative humidity scale (water uptake starts at 71% for SSA and 75% RH for NaCl). The dashed orange line in Fig. 6a therefore only denotes the starting point for the strong water uptake by the SSA particles, but the deliquescence is not yet completed and heterogeneous ice formation can still proceed; we unfortunately missed to describe this more clearly in the manuscript. The deliquescence is only completed at a higher RH value, which as outlined above, would be obtained by scaling the extrapolated, temperature-dependent parameterisation of the deliquescence relative humidities of crystalline NaCl particles from Tang and Munkelwitz (1993) with an absolute, temperature-independent shift of –1% on the relative humidity scale. In the revised versions of Figs. 6 & 7, we therefore replaced the dashed orange line by an orange shaded area that comprises the RH *range* of the full deliquescence step of the SSA particles. One can then see in Fig. 6a that the maximum of the fractional ice activation of the SSA particles still lies with that range. The calculation of the orange shaded area is described on page 17, line 535:

*"The estimated RH range of the full deliquescence (FDRH) of inorganic sea salt particles is indicated by the orange shaded area. At 298 K, Tang et al. (1997) observed the full dissolution of levitated sea salt aerosol particles at RH between 71 and 74% RH, whereas pure NaCl particles deliquesced at 75.3% RH. We therefore scaled the extrapolated, temperature-dependent parameterisation of the deliquescence relative humidities of pure NaCl particles from Tang and Munkelwitz (1993) with an absolute, temperature-independent shift between –4% and –1% on the relative humidity scale to estimate the RH range for the full dissolution of the inorganic sea salt particles at low temperatures."*

Added references:

Tang, I. N., Tridico, A. C., and Fung, K. H.: Thermodynamic and optical properties of sea salt aerosols, J. Geophys. Res. (Atmos.), 102, 23269-23275, 1997.

Tang, I. N., and Munkelwitz, H. R.: Composition and Temperature-Dependence of the Deliquescence Properties of Hygroscopic Aerosols, Atmos. Env., 27A, 467-473, 10.1016/0960-1686(93)90204-C, 1993.

3. The differences between the present study's findings and those from Wilson et al. 2015 and Wolf et al. 2020 are interesting. I would like to see a little more exploration of the oceanographic reasons as to why the results from this study (from the Arctic) do not follow the same patterns as those from Wilson et al. and Wolf et al (from the tropics).

a. For example: were the average rates of primary productivity, and therefore perhaps the organic content of the seawater, higher in the tropics? And if so, could this help to explain the observed higher nS values for the seawater from Wilson et al.'s and Wolf et al.'s analyses? This could be easily explored by comparing satellite retrievals of surface chlorophyl-a concentrations.

We currently cannot convincingly explain the discrepancy between our findings and those from Wilson et al. (2015) and Wolf et al. (2020), but of course, we agree to expand our discussion on the possible link between the local biological activity and the organic carbon enrichment in the SSA particles, which could affect their ice nucleation ability. There is still a controversial debate as to which extent the organic enrichment in sea spray aerosol is controlled by the primary productivity in marine environments, which is conveniently characterised by chlorophyll-*a* levels of seawater as a measure of phytoplankton biomass. On the one hand, several studies have found that the organic matter enrichment in sea spray is directly linked to primary production (e.g. Ceburnis et al., 2011, 2016; van Pinxteren et al., 2017), yielding higher enrichment factors of organic carbon in periods of high chlorophyll-*a* concentrations. On the other hand, it was argued that the local biological activity, as measured by chlorophyll-*a*, is of minor importance and uncoupled from a large reservoir of organic carbon in ocean surface waters, which primarily controls the enrichment of organic matter in SSA particles (Quinn et al., 2014). Recent cruises in the North Atlantic Ocean also showed that the size-resolved organic mass fractions and the CCN activity of in situ generated SSA particles were relatively invariant although the sampling regions featured a wide range in phytoplankton biomass (chlorophyll-*a* concentrations from 0.1 to > 2.0 mg m$^{-3}$) and a broad diversity of phytoplankton components (Bates et al., 2020).

Wolf et al. (2020) have argued that the higher ice nucleation ability of particles generated from the sea surface microlayer samples in the Eastern Tropical North Pacific (ETNP) Ocean compared to those in the Florida Straits is linked to primary productivity. As one indicator, the average surface chlorophyll-*a* concentrations from satellite retrievals were higher for the ETNP sampling location compared to the Florida straits (0.19 vs. 0.10 mg m$^{-3}$). However, there was no direct correlation between the chlorophyll-*a* concentrations and the critical ice saturation

ratio for the individual samples, meaning that highest chlorophyll-*a* concentrations did not induce heterogeneous ice formation at the lowest critical ice saturation ratio. For immersion freezing measurements under mixed-phase cloud conditions, Irish et al. (2019) also did not find a statistically significant correlation between the $T_{10}$ freezing temperatures (corresponding to a frozen fraction of 10%) and the chlorophyll-*a* concentrations of the investigated microlayer samples from the Canadian Arctic. As described in our article, we have used a subset of these samples for the present AIDA ice nucleation experiments under cirrus conditions. The satellite-retrieved chlorophyll-*a* concentrations for the sampling locations in the Canadian Arctic ranged between 0.5 and 1.4 mg m$^{-3}$ (see Fig. 7 in Irish et al., 2019), thus being higher than the surface chlorophyll-*a* concentrations for the ETNP and Florida Straits summarized in Table 1 of Wolf et al. (2020). However, the AIDA data revealed a poorer heterogeneous ice nucleation ability of the particles generated from the Canadian Arctic samples in comparison with the CFDC measurements from Wolf et al. (2020) with particles generated from the ETNP and Florida Straits samples. This indicates that factors other than primary productivity are important for explaining the difference in the SSA particles' ice nucleation ability. One of those factors could be the biogeographic pattern of the phytoplankton species. The laboratory studies summarised in Sect. 1.2 of our article underline that there are notable variations in the heterogeneous ice nucleation ability of various phytoplankton species under cirrus conditions, with e.g. *Prochlorococcus* showing distinctly lower critical ice saturation ratios compared to *Thalassiosira pseudonana*, *Nannochloris atomus*, and *Emiliania huxleyi.* The phytoplankton species richness in the tropics was found to be about three times that in higher latitudes (Righetti et al., 2019), potentially increasing the probability that a particularly ice-active species can be found in field-collected microlayer samples from the tropics. *Melosira arctica*, the most productive algae in the Arctic Ocean (Booth and Horner, 1997), was not a source of particularly active INPs in our previous AIDA ice nucleation measurements that focussed on the mixed-phase cloud temperature region (Ickes et al., 2020).

We will add some of these aspects into Sect. 4.2 to extend our previous discussion on page 19, line 608 as follows:

[revised manuscript text omitted]

4. The authors propose that the ice nucleation measurement technique might contribute to the differences between their results and the results of Wilson et al. and Wolf et al. Specifically, the authors suggest that the differences in temperature and relative humidity trajectories experienced by aerosol particles in the AIDA and CFDC techniques may impact ice nucleation behaviour. The discussion covering this (Section 4.4) is rather speculative. It could be strengthened with further citations and details on key points the authors touch on. I outline a few specific recommendations below:

Thank you for these considerations and suggestions. We will include them in the revised version of our article as outlined below.

a. The authors should better reference the literature on temperature and relative humidity trajectories as particles enter CFDCs. For example, I recommend citing and discussing the results in Rogers, 1988, Garimella et al., 2016, and Kulkarni and Kok, 2012. These papers include simulations describing the T and RH trajectories experienced by particles as they enter the CFDC.

Based on your suggestion, we will extend our previous, brief discussion of the temperature and RH trajectory in the CFDCs on page 21, line 671 as follows:

 "*The aerosol is surrounded by two sheath air flows, so that the sample temperature and the water vapour environment are very narrow and well defined (Rogers, 1988). The location of the aerosol lamina and its associated temperature and supersaturation conditions can be accurately calculated from instrumental parameters such as wall temperatures, sheath flow rates, and sample flow rates (Rogers, 1988; Kulkarni and Kok, 2012; Garimella et al., 2016). Computational fluid dynamics calculations show that the initially warm and dry sample air flow quickly adopts the nominal lamina temperature and $S_{ice}$ value within the upper 5–10% section of the main chamber (Garimella et al., 2016). As such, an aerosol particle in the centre of the flow region of a CFDC at $S_{ice}$ 1.3 and 225 K is almost instantly subjected to these ice supersaturated conditions after drying and has not experienced the same RH history as in the AIDA chamber.*"

Added references:

Garimella, S., Kristensen, T. B., Ignatius, K., Welti, A., Voigtlander, J., Kulkarni, G. R., Sagan, F., Kok, G. L., Dorsey, J., Nichman, L., Rothenberg, D. A., Rosch, M., Kirchgassner, A. C. R., Ladkin, R., Wex, H., Wilson, T. W., Ladino, L. A., Abbatt, J. P. D., Stetzer, O., Lohmann, U., Stratmann, F., and Cziczo, D. J.: The SPectrometer for Ice Nuclei (SPIN): an instrument to investigate ice nucleation, Atmos. Meas. Tech., 9, 2781-2795, 10.5194/amt-9-2781-2016, 2016.

Kulkarni, G. and Kok, G.: Mobile Ice Nucleus Spectrometer, Technical Report No. PNNL-21384, Pacific Northwest National Lab. (PNNL), Richland, WA, US, 2012.

Rogers, D. C.: Development of a continuous flow thermal gradient diffusion chamber for ice nucleation studies, Atmos. Res., 22, 149-181, 1988.

b. Once the T and RH trajectories are better discussed, the authors should expand on how the differences between CFDCs and AIDA can lead to physical differences in particle water uptake. The authors should mention that the rates of water uptake through organic coatings can be estimated for model organics (e.g. Price et al., 2015; Renbaum-Wolff et al., 2013). These calculations, even if not yet possible for complex mixtures of marine organics, at least provide a theoretical underpinning for the types of issues the authors raise.

Yes – we have already briefly mentioned the modelling studies that investigated the kinetic limitations of water diffusion into the particles in the introduction on page 7 (lines 221 – 223), but it is certainly useful to refer to them again in Sect 4.4. On page 22, line 677, we stated: "*Organic-rich particles might prevail in a highly viscous or glassy state at low temperature, with the result that there is a competition between water uptake and deposition ice nucleation on the glassy, solidified organic surface (Reid et al., 2018).*" We propose to add here the following paragraph, where we again refer to the modelling studies and describe one exemplary finding:

*"The effect of kinetic limitations of water diffusion and its impact on equilibration timescales and modes of ice nucleation have already been investigated in various computational studies with model organic substances (e.g. Berkemeier et al., 2014; Lienhard et al., 2015; Price et al., 2015; Fowler et al., 2020). For example, Price et al. (2015) have modelled equilibration times for $\alpha$-pinene secondary organic material based on experimental diffusion measurements. At temperatures of 260 K and above, these timescales were faster than 1 s for the considered RH range between 5 and 95%. At 240 K, the response time was already in the range of a couple of seconds for low RH values, and might further increase up to hours at upper-tropospheric temperatures (Price et al., 2015)."*

c. The authors might expand on their call for an intercomparison by suggesting that part of the intercomparison could be to subject aerosols to various "precooling" trajectories, i.e., controlling the temperature and relative humidity of aerosols in a large mixing chamber

(NAUA?), prior to CFDC sampling. This could help facilitate comparison between AIDA and CFDC data.

This is indeed an important aspect. Ladino et al. (2014) have shown that precooling is a factor that controls the ice nucleation ability of highly viscous organic aerosol particles. Precooling led to a decrease in the particles' ice nucleation onsets, presumably because the particles were more viscous or solid-like. Our smaller aerosol preparation chamber NAUA can be operated at room temperature, so it would be feasible to do the CFDC sampling from the NAUA chamber at 298 K and various RH conditions and compare these measurements to those from low-temperature particle sampling from the AIDA chamber. We will add this point to our discussion on page 22, line 693:

"*Moreover, Ladino et al. (2014) have shown that precooling is a factor that controls the ice nucleation ability of highly viscous organic aerosol particles. Precooling led to a decrease in the particles' ice nucleation onsets, presumably because the particles were more viscous or solid-like. Another subject of the proposed intercomparison could be exposing the particles to various RH and temperature conditions prior to CFDC sampling and examining the associated effect on their ability to nucleate ice.*"

New reference:

Ladino, L. A., Zhou, S., Yakobi-Hancock, J. D., Aljawhary, D., and Abbatt, J. P. D.: Factors controlling the ice nucleating abilities of alpha-pinene SOA particles, J. Geophys. Res. (Atmos.), 119, 9041-9051, 10.1002/2014jd021578, 2014.

d. The manuscript is already quite long, and I do not mean to require that the authors add much text in response to the above points. Two or three sentences for each point should suffice.

Yes, we have tried to formulate short paragraphs to address the individual points.

**Minor Comments**

**Abstract**

5. Line 16: "*Only a small fraction of sea salt aerosol is transported to the upper troposphere...*" Please change "sea salt aerosol" to "sea spray aerosol," since these aerosol particles are often internally mixed and compositionally complex, consisting of more than just salts.

Yes, will be changed as suggested.

6. Line 22: "*The particles were suspended in a large cloud chamber...*" I think it would be useful to specifically mention AIDA here.

Yes, will be changed to *"… were suspended in the AIDA cloud chamber …"*.

7. Line 32: "*we also discuss how far instrumental parameters...*" Semantical point, but I feel you don't discuss "har far" – i.e., quantify the extent to which – these parameters might impact results. You only discuss that they could impact the results without supplying an estimate for how large the magnitude of the impact might be. I would remove the word "far."

Agreed – we will remove the word "far".

**Introduction**

8. Line 44: "*…homogeneous freezing of pure water droplets, which takes place below about 235 K.*" Please cite Koop et al. 2000 here, or your reference of choice.

Yes, we will cite Koop et al. (2000b) from our reference list.

9. Line 63: "*…the freezing data are usually reported as the temperature-dependent number of INPs per either droplet volume or volume of collected air.*" Temperature-dependence is reported for INP concentrations in the mixed-phase cloud regime; but in the cirrus regime below liquid water saturation, INP concentration is reported as a function of both temperature *and* relative humidity. Please clarify this in the text.

Good point, we will add the following sentence on line 64:

*"For ice nucleation measurements under cirrus conditions (see Sect. 1.2), INP concentrations are reported as a function of temperature and relative humidity."*

10. Line 78: "*…showed contributions of up to 25% from sea salt over ocean regions.*" Please change to "over ocean and coastal regions."

Yes, will be changed accordingly.

11. Line 81: "*…if we are to explain regional indications of heterogeneous ice nucleation activity...*" I'm not sure I understand the meaning of the word "indications" here. Perhaps change to "importance" or "impact" or "variability?"

Yes, "importance" is a better word here, will be changed accordingly.

12. Lines 155-158: Can the authors clarify what is meant by "partial deliquescence" or "before full deliquescence?" Do they refer to pre-deliquescence uptake of water (i.e., hygroscopic growth below the DRH)? Or do they refer to the time between the start of deliquescence and full deliquescence above the DRH?

For these multicomponent hygroscopic aerosol particles, the terms "partial deliquescence" and "before full deliquescence" denote the behaviour that the particles go through partially dissolved states before finally becoming a homogeneous solution droplet. Dry sea salt aerosol particles begin to deliquesce at a low RH due to the presence of certain salts ($KMgCl_3 \cdot 6H_2O$, $MgCl_2$) of low deliquescence RH. Only at about 74% RH (at 298 K), all of the remaining NaCl

has dissolved and the particle finally becomes a homogeneous aqueous solution droplet (Tang et al., 1997). On line 150, we stated: *"In the case of SSA particles that contain not just a single but a mixture of inorganic salts, deliquescence is a gradual process."* We propose to extend this description as follows:

*"These particles go through partially dissolved states before finally becoming homogeneous aqueous solution droplets. They begin to deliquesce at a low RH due to the presence of Ca and Mg salts with low deliquescence points, but only at about 74% RH (298 K), all of the remaining NaCl is dissolved and the particles transform to homogeneous droplets (Tang et al., 1997)."*

Tang, I. N., Tridico, A. C., and Fung, K. H.: Thermodynamic and optical properties of sea salt aerosols, J. Geophys. Res. (Atmos.), 102, 23269-23275, 1997.

13. Line 166 and throughout: I think "*Emiliana huxleyi*" should be spelled "*Emiliania huxleyi*."

14. Line 174 and throughout: I think "*Perchlorococcus*" should be spelled "**Pro***chlorococcus.*"

Thank you very much for spotting these two errors. We apologise for the negligence and correct all misspellings.

**Experimental**

15. Line 283: "…*particles generated by nebulising the undiluted microlayer and surface seawater samples…*" were the samples homogenized (e.g., shaken) after thawing and prior to aerosolization?

Yes, the samples were shaken – we will add this information to the first sentence of Sect. 2.2 (line 269):

*"For the ice nucleation measurements in the AIDA chamber, the seawater samples and the SM100 cultures were thawed, homogenised by shaking, and aerosolised with an ultrasonic nebuliser (GA2400, SinapTec)."*

16. Line 318: *"For a subset of samples, we diluted the suspensions by a factor of 10 and 100 with ultrapure water to extend the measured nINP(T) spectrum to lower freezing temperatures.*" At what temperatures was this extra dilution step necessary?

The FF curves depicted in Fig. 2a show that around half of the undiluted samples were already completely frozen above 260 K. The accessible temperature range for the INSEKT measurements, however, extends down to about 251 K, where the FF curve of the blank measurement with ultrapure water starts to increase. The dilution factor also increases the range of cumulative INP concentrations (see Eq. 3 in Ickes et al., 2020), so that our measurements with a comparatively large aliquot volume of 50 $\mu$l have a better overlap with

those in which a smaller aliquot volume of 1 μl was used (see our Fig. 2b). We will add this aspect to the sentence above:

"*For a subset of samples, we diluted the suspensions by a factor of 10 and 100 with ultrapure water to extend the measured $n_{INP}(T)$ spectrum to lower freezing temperatures and higher cumulative INP concentrations. Thereby, we achieved a better overlap with other cold-stage measurements in which smaller aliquot volumes of 1 μL were used (Wilson et al., 2015; Irish et al. 2019).*"

17. Line 342: *"…we calculated the ice nucleation active surface site density, ns, with an estimated uncertainty of ± 40%.*"

This uncertainty range seems rather large. Can the authors briefly summarize here the factors that go into calculating uncertainty in nS?

Ullrich et al. (2017), given by us as the reference for the estimated uncertainty, have derived the ±40% value using error propagation with individual uncertainties of ±20% for $N_{ice}$ (measurement uncertainty of the optical particle counters) as well as ± 34% for the total aerosol particle surface area concentration, $N_{aer} \cdot A_{aer}$. The latter value considers the measurement uncertainty for $N_{aer}$ and the uncertainties in the determination of the average surface area for aspherical particles, where particle density and shape factor are needed to convert the measured mobility and aerodynamic diameters into volume-equivalent sphere diameters (see line 1195ff, caption of Fig. 1). We will extend our statement on line 342 as follows:

"*The uncertainty of $n_s$ was estimated to ±40%, using error propagation with individual uncertainties of ± 20% for $N_{ice}$ as well as ± 34% for $N_{aer} \cdot A_{aer}$ (Ullrich et al., 2017).*"

18. Line 345: *"…we consider the extreme scenario…"* Is this considered "extreme" because experimental experience demonstrates that counting frequency is typically much higher?

"Extreme" is intended to refer to the lower detection limit with respect to the heterogeneous ice nucleation mode, i.e., that there would only be one heterogeneously nucleated ice crystal in the considered time period. We will replace the term *"extreme scenario"* by *"lower limit case"*.

19. Lines 346-349: This is a nice description of lower limits!

Thank you!

**Results**

20. Line 362: *"…corrected for the freezing point depression by the salts.*" Can the authors provide more description, or a reference, as to how this was freezing point depression correction was done?

We have already provided these references, which describe the freezing point depression correction in detail, in Sect. 2.2, line 313-314. To emphasise this, we will add the term *"(see Sect. 2.2)"* at the end of line 362.

21. Line 373: "*This may be partly explained by the weather conditions...*" E.g., high winds? Can the author provide a typical wind speed during these measurements? The authors might reference one of the numerous studies indicating the wind speed at which the microlayer breaks up, e.g. Wurl et al., 2011.

Thank you for pointing to this issue. "Weather conditions" is indeed an unprecise statement. Typical wind speeds during sampling at the Kongsfjorden site are given in Appendix A (line 752; they ranged between 3.4 and 6.2 m/s). These values are smaller than those for which a disruption of the sea surface microlayer would be expected (see e.g. Wurl et al., 2011). But we have described in Sect. 2.1 that the seawater samples from Kongsfjorden were not microlayer samples but "surface seawater samples" due to the employed sampling technique (Niskin sampler placed horizontally on the water surface). What we meant by "weather conditions" is that the (relatively) rough sea conditions also made the horizontal placement of the Niskin sampler more difficult, so that we concluded in line 248 that the "*material will have been heavily diluted with subsurface water*". So we did not intend to refer to the disruption of the microlayer. Rather, we wanted to suggest that the strong dilution with subsurface water could have contributed to a homogenisation of the KFJ surface seawater samples in comparison with the higher degree of variability (in the ice nucleation ability) observed for the SML and STN microlayer samples.

So we will clarify our statement at line 373 as follows:

*"This may be partly explained by the strong dilution with subsurface waters, leading to a homogenisation of the KFJ surface seawater samples in comparison with the higher degree of variability observed for the SML and STN microlayer samples."*

22. Line 409: "*…any observable heterogeneous ice nucleation mode must be related to the organic material contained in the aerosol particles because the inorganic salt components are not yet ice-active at this temperature.*" Is it not also possible that the seawater also contains dust? See e.g. Cornwell et al., 2020.

Yes, Cornwell et al. (2020) have suggested that re-suspended dust should be considered as another possible source of ocean-emitted INPs; we have mentioned this study in our introduction (lines 51-53). We propose to change our above statement to:

*"… must be related to organic material or other ice-nucleating entities like dust contained in the aerosol particles …"*

**Discussion and Outlook**

23. Line 574: "…the processing of exudates either through biological processes such as microbial metabolism or physicochemical processes…" Please add a reference to support this discussion of microbial metabolism. I suggest either McCluskey et al., 2017 and/or Wang et al., 2015, but feel free to add another.

Yes, good suggestion, we will add both McCluskey et al. (2017) and Wang et al. (2015) as references.

24. Line 583: "However, the amount of dispersed ice-nucleating entities was obviously much smaller than in the Wolf et al. (2019) study." Please state whether the cell concentrations were similar or different between this study and Wolf et al.

Thank you for raising this point. It is indeed worthwhile to highlight here the different cell concentrations in the Wolf et al. (2019) study ($5 \cdot 10^8$ cells/mL of *Prochlorococcus*, see line 176) and in our work ($2.85 \cdot 10^6$ cells/mL of *Skeletonema marinoi,* see line 263). We will therefore add the following statement:

"This could be due to the different cell concentrations of the suspensions examined in our work compared to those in Wolf et al. (2019), i.e., $2.85 \cdot 10^6$ cells/mL of *Skeletonema marinoi* vs. $5 \cdot 10^8$ cells/mL of *Prochlorococcus*."

25. Line 628: "…where the bursting of bubble cap films can lead to the formation of highly organically enriched particles." Please add a reference.

Yes, we will add again the three references from line 51 where we already addressed this issue, i.e., O'Dowd et al., 2004; Ault et al., 2013; and Prather et al., 2013.

26. Line 631: *"…the ice nucleation mode might change from immersion freezing, as observed in the AIDA experiments, to deposition nucleation, where ice formation initiates by the deposition of water vapour on crystalline or glassy surfaces."*

Please add a reference (or two) that discusses depositional freezing on glassy organic aerosols. E.g. Murray et al., 2010.

Yes, good point – we will add Murray et al. (2010) and Wilson et al. (2012):

Murray, B. J., Wilson, T. W., Dobbie, S., Cui, Z., Al-Jumur, S. M. R. K., Möhler, O., Schnaiter, M., Wagner, R., Benz, S., Niemand, M., Saathoff, H., Ebert, V., Wagner, S., and Kärcher, B.: Heterogeneous nucleation of ice particles on glassy aerosols under cirrus conditions, Nature Geoscience, 3, 233-237, 2010.

Wilson, T. W., Murray, B. J., Wagner, R., Möhler, O., Saathoff, H., Schnaiter, M., Skrotzki, J., Price, H. C., Malkin, T. L., Dobbie, S., and Al-Jumur, S. M. R. K.: Glassy aerosols with a range of compositions nucleate ice heterogeneously at cirrus temperature, Atmos. Chem. Phys., 12, 8611–8632, 2012.

27. Line 673: "Its transit time through the nucleation region of a CFDC is typically about 10 seconds."

Please add a reference for this residence time. E.g. Garimella et al., 2016.

Yes, we suggest adding Rogers (1988) as residence time is discussed more explicitly in this article.

**Figures and Tables**

Table 1: Be sure to correct the spelling of the names here, as indicated above.

Thanks again for spotting these errors, we will correct all misspellings.

Table 2: What does the uncertainty or variability in the mean diameter (±0.05 μm) represent?

This value denotes the approximate bin width of the size channels of the APS instrument in the size range from 0.6 – 0.9 μm where the main particle mode is located. As this is not immediately related to the uncertainty estimate for $n_s$, which we have now outlined in more detail in our answer to point 17 above, we suggest deleting this bin width value to avoid any confusion.

---

## Author Comment (AC2)

**Response to Referee #2 (Luis Antonio Ladino)**

We thank Luis Antonio Ladino for the positive evaluation of our article. Below, we provide a point-by-point answer to his comments (referee report in blue, our answers in black). The page and line numbers refer to the original manuscript.

General comment:

In the present study the ice nucleation abilities of marine aerosol particles relevant to mixed-phase and cirrus clouds are presented based on previous observations and a new set of experiments. The results and the conclusions from the present study are a great contribution to the ice nucleation community as it helps us to improve the current understanding that marine aerosol particles play in cloud formation. This is a well designed and executed study where the authors paid a lot of attention to each experiment to properly interpret it. The manuscript is very well written with a sound discussion where the potential sources of uncertainties are highlighted and described. The manuscript can basically be accepted as is. However, below five minor comments are included to be considered in the final manuscript.

Minor comments:

Line 44: Add a reference after "235 K".

Yes – as also suggested by Referee #1, we will cite Koop et al. (2000b) from our reference list.

Line 119: What do the authors mean with "constant composition"?

This refers to the fact that the amount of water vapour within the sealed environmental cell is small compared to the amount of liquid water in the aqueous NaCl solution droplets (with immersed diatom cells). The composition of the solution droplets, i.e., the weight fraction of NaCl, therefore remains constant during cooling because condensation of the available water vapour is negligible compared to the condensed phase water in the droplets. We suggest deleting *"at constant composition"* from line 119 and adding another sentence to explain this in more detail:

*"The composition of the NaCl solution droplets remained constant during cooling because the amount of water vapour in the environmental cell was negligible compared to the amount of liquid water in the droplets."*

Line 181: "In particular smaller, 200 nm-sized particles showed". This does not read properly.

We propose to rephrase this sentence as follows:

*"The increase in the organic carbon content after cell lysis was particularly pronounced for smaller particle sizes (200 nm). These 200 nm-sized particles proved to be very efficient INPs,*

*with ice-active fractions > 1% at $S_{ice}$ >1.18 and corresponding $n_s$ densities that were similar in magnitude to those of other common INPs like mineral and soil dust (Wolf et al., 2019)."*

Lines 442-444: Please double check if the sea surface microlayer samples during the ACCACIA expedition were indeed collected using a glass plate.

Thank you for spotting this. The samples during ACCACIA were actually collected from a hydrophilic Teflon film on a rotating drum. We will correct our sentence on line 242 as follows:

*"The sea surface microlayer samples from the Eastern Canadian Arctic and the Greenland Sea were collected with the glass plate technique during NETCARE (Irish et al., 2019) and from a hydrophilic Teflon film on a rotating drum during ACCACIA (Wilson et al., 2015) field expeditions …"*

Table A1. I do not see the purpose of adding it to the manuscript.

We acknowledge that our discussion is not strongly linked to this table. In the interests of complete documentation, we would still like to include the data. There might be future studies on the ice nucleation ability of sea spray aerosol particles under cirrus conditions (with other field-collected microlayer samples), for which our tabulated data for e.g. dissolved organic carbon (DOC) and bacterial abundance ($N_{bac}$) could be a valuable reference, as these parameters could affect the observed ice nucleation behaviour.